# SPECIAL UNITARY PARAMETERIZED ESTIMATORS OF ROTATION

**Akshay Chandrasekhar**

## ABSTRACT

This paper revisits the topic of rotation estimation through the lens of special unitary matrices. We begin by reformulating Wahba's problem using $SU(2)$ to derive multiple solutions that yield linear constraints on corresponding quaternion parameters. We then explore applications of these constraints by formulating efficient methods for related problems. Finally, from this theoretical foundation, we propose two novel continuous representations for learning rotations in neural networks. Extensive experiments validate the effectiveness of the proposed methods.

## 1 INTRODUCTION

3D rotations are fundamental objects ubiquitously encountered in domains such as physics, aerospace, and robotics. Many representations have been developed over the years to describe them including rotation matrices, Euler angles, and quaternions. Each method has specific strengths such as parameter efficiency, singularity avoidance, or interpretability. While special orthogonal matrices $SO(3)$ are widely used, their complex counterparts, special unitary matrices $SU(2)$, are less explored in areas like robotics and machine learning. This paper showcases the utility of special unitary matrices by tackling rotation estimation from different perspectives.

### 1.1 WAHBA'S PROBLEM

Wahba's problem (Wahba, 1965) is a fundamental problem in attitude estimation. The task refers to the process of determining the orientation of a target coordinate frame relative to a reference coordinate frame based on 3D unit vector observations. More formally, it is phrased as seeking the optimal rotation matrix $\mathbf{R}$ minimizing the following loss:

$$\min_{\mathbf{R} \in SO(3)} \sum_i w_i ||\mathbf{b}_i - \mathbf{R}\mathbf{a}_i||^2 \tag{1}$$

where $\mathbf{a}_i$ are the reference frame observations, $\mathbf{b}_i$ are the corresponding target frame observations, and $w_i$ are the real positive weights for each observation pair. The problem can be solved analytically by finding the nearest special orthogonal matrix (in a Frobenius sense) to the matrix $\mathbf{B}$ below:

$$\mathbf{B} = \sum_i w_i \mathbf{b}_i \mathbf{a}_i^T \tag{2}$$

Today, this solution is typically computed via singular value decomposition (Markley, 1987).

Alternatively, the solution can be estimated as a unit quaternion. Davenport (1968) introduced the first such method in 1968 by showing that the optimal quaternion $\mathbf{q}$ is the eigenvector corresponding to the largest eigenvalue of a 4x4 symmetric gain matrix $\mathbf{K}$, which can be constructed as:

$$\mathbf{K} = \begin{bmatrix} Tr(\mathbf{B}) & \mathbf{z}^T \\ \mathbf{z} & \mathbf{B} + \mathbf{B}^T - Tr(\mathbf{B})\mathbf{I} \end{bmatrix} \tag{3}$$

where $\mathbf{I}$ is the identity matrix, $Tr(\mathbf{B}) = \sum_i \mathbf{B}_{ii}$, and $\mathbf{z} = \sum_i w_i \mathbf{a}_i \times \mathbf{b}_i$. The solution via eigendecomposition is relatively slow as it solves for all the eigenvectors of the matrix which are not needed. Later solutions improve upon this by calculating the characteristic equation of $\mathbf{K}$ and solving for only the largest eigenvalue (Shuster and Oh, 1981; Mortari, 1997; Wu et al., 2018). For an overview of major algorithms, see Lourakis and Terzakis (2018).

## 1.2 Representations for Learning Rotations

In recent years, there has been great interest in representing rotations within neural networks, which often struggle with learning structured outputs. Directly predicting common parameterizations such as quaternions or Euler angles has generally performed relatively poorly (Geist et al., 2024). In fact, it was shown that any 3D rotation parameterization in less than five real dimensions is discontinuous, necessitating non-minimal representations for smooth learning (Zhou et al., 2019). Additionally, challenges like double cover in some representations can further hinder learning. Two leading approaches, Levinson et al. (2020) and Peretroukhin et al. (2020), essentially interpret network outputs as $\mathbf{B}$ and $\mathbf{K}$ matrices (Eqs. (2) and (3) respectively), mapping them to rotations via solutions to Wahba's problem. Thus, the two tasks can be linked. For a more in depth overview of the task, see Geist et al. (2024).

## 1.3 Contributions

The growing link between solutions to Wahba's problem and effective rotation representations for learning motivates a deeper re-examination of the classic problem. In this work, we derive several new solutions to Wahba's problem using the structure of special unitary matrices and discuss how these solutions can be leveraged in classical rotation optimization. From these solutions, we introduce two new representations for learning, 2-vec and QuadMobius. We demonstrate through experiments that 2-vec outperforms the comparable Gram-Schmidt representation (Zhou et al., 2019) given the same compute and dimensionality, and QuadMobius achieves state-of-the-art performance across different tasks. Detailed proofs, derivations, and theoretical experiments are provided in the Appendix to further support our contributions.

**We highly recommend that the reader first review Appendix A to become familiar with the relevant mathematical background and notation used throughout the paper.** Reference code for proposed methods is available at https://github.com/akschion/SUPER.

## 2 Solutions to Wahba's Problem via SU(2)

Transferring Wahba's Problem to complex projective space, we can solve for the optimal rotation through special unitary matrices.

## 2.1 Stereographic Plane Solution

First, we establish the proper distance metric in complex projective space corresponding to the spherical chordal metric in Eq. (1). For points $\mathbf{a}, \mathbf{b} \in S^2$ and their stereographic projections $\psi(\mathbf{a}) = \mathbf{z} = [z_1, z_2]^T$ and $\psi(\mathbf{b}) = \mathbf{p} = [p_1, p_2]^T$, we can show that the metric can be expressed in the following way (derivation in Appendix B.1.1):

$$||\mathbf{a} - \mathbf{b}||^2 = \frac{4|z_1 p_2 - z_2 p_1|^2}{||\mathbf{z}||^2 ||\mathbf{p}||^2} \tag{4}$$

We now seek to find the rotation $\mathbf{R}$ parameterized by corresponding special unitary matrix $\mathbf{U}$ in complex projective space that minimizes the objective in Eq. (1). Applying our derived metric and Eqs. (32) and (34), we can construct for each weighted input correspondence $\mathbf{z}_i$ and $\mathbf{p}_i$:

$$w_i ||\mathbf{b}_i - \mathbf{R}\mathbf{a}_i||^2 = \frac{4w_i|(-\bar{\beta}z_{i,1} + \bar{\alpha}z_{i,2})p_{i,1} - (\alpha z_{i,1} + \beta z_{i,2})p_{i,2}|^2}{(|\alpha z_{i,1} + \beta z_{2,i}|^2 + |-\bar{\beta}z_{1,1} + \bar{\alpha}z_{i,2}|^2)||\mathbf{p}_i||^2}$$

$$= \frac{4w_i|(-\bar{\beta}z_{i,1} + \bar{\alpha}z_{i,2})p_{i,1} - (\alpha z_{i,1} + \beta z_{i,2})p_{i,2}|^2}{||\mathbf{U}\mathbf{z}_i||^2 ||\mathbf{p}_i||^2}$$

where $\alpha, \beta$ are the complex parameters defining $\mathbf{U}$ from Eq. (31). By definition of unitary matrices, $||\mathbf{U}\mathbf{z}||^2 = ||\mathbf{z}||^2$. Thus, we can rewrite our expression as the following target constraint:

$$\frac{4w|(-\bar{\beta}z_{i,1} + \bar{\alpha}z_{i,2})p_{i,1} - (\alpha z_{i,1} + \beta z_{i,2})p_{i,2}|^2}{||\mathbf{z}||^2||\mathbf{p}||^2} = 0 \tag{5}$$

$$\implies \frac{2\sqrt{w}((-\bar{\beta}z_{i,1} + \bar{\alpha}z_{i,2})p_{i,1} - (\alpha z_{i,1} + \beta z_{i,2})p_{i,2})}{\sqrt{|z_{i,1}|^2 + |z_{i,2}|^2}\sqrt{|p_{i,1}|^2 + |p_{i,2}|^2}} = 0 \tag{6}$$

The expression is now just a linear function of rotation parameters. It is a general constraint as it handles the entire complex projective space (proof in Appendix B.1.2). However, in practice, our inputs are more commonly given as projection coordinates on the complex plane. As such, we have:

$$z_{i,1} = z_i = x_i + y_i i, \quad p_{i,1} = p_i = m_i + n_i i, \quad z_{i,2} = p_{i,2} = 1$$

for each point correspondence ($x_i, y_i, m_i, n_i \in \mathbb{R}$). This simplifies the constraint to the following:

$$\frac{2\sqrt{w_i}((-\bar{\beta}z_i + \bar{\alpha})p_i - \alpha z_i - \beta)}{\sqrt{|z_i|^2 + 1}\sqrt{|p_i|^2 + 1}} = 0 \tag{7}$$

We can rearrange the equation to the following linear form with $\mathbf{u} = \begin{bmatrix} \alpha & \beta & \bar{\alpha} & \bar{\beta} \end{bmatrix}^T$:

$$w_i' = \frac{4w_i}{(|z_i|^2 + 1)(|p_i|^2 + 1)} \tag{8}$$

$$\sqrt{w_i'} \begin{bmatrix} -z_i & -1 & p_i & -p_i z_i \end{bmatrix} \mathbf{u} = \sqrt{w_i'}\mathbf{A}_i\mathbf{u} = 0 \tag{9}$$

Each input point pair gives us a complex constraint $\mathbf{A}_i$. Stacking $\mathbf{A}_i$ together and multiplying the weights through, we can write the relation succinctly as $\mathbf{A}\mathbf{u} = 0$ ($\mathbf{A}$ is a complex $n$ x 4 matrix for $n$ points). With noisy observations, the constraints do not hold exactly, so we aim to find the best rotation that minimizes the least squares error $||\mathbf{A}\mathbf{u}||^2$. It is nontrivial to solve for the minimizing vector $\mathbf{u}$ while ensuring the result will form a valid special unitary matrix ($\mathbf{u}_1 = \bar{\mathbf{u}}_3$, $\mathbf{u}_2 = \bar{\mathbf{u}}_4$, $\mathbf{u}_1\bar{\mathbf{u}}_1 + \mathbf{u}_2\bar{\mathbf{u}}_2 = 1$). To more effectively solve this, we use Eq. (35) to transform the vector $\mathbf{u}$ to a corresponding quaternion $\mathbf{q} = \begin{bmatrix} w_q & x_q & y_q & z_q \end{bmatrix}^T$ that has a simpler constraint ($\mathbf{q}$ must be unit norm). We carry out the complex multiplication for each $\mathbf{A}_i\mathbf{u}$ and break the constraint into two constraints, one for the real and imaginary parts respectively:

$$w_i' = \frac{4w_i}{(1 + x_i^2 + y_i^2)(1 + m_i^2 + n_i^2)} \tag{10}$$

$$\sqrt{w_i'} \begin{bmatrix} x_i - m_i & -y_i - n_i & 1 + m_i x_i - n_i y_i & m_i y_i + n_i x_i \\ y_i - n_i & x_i + m_i & m_i y_i + n_i x_i & 1 - m_i x_i + n_i y_i \end{bmatrix} \mathbf{q} = \sqrt{w_i'}\mathbf{D}_i\mathbf{q} = 0 \tag{11}$$

Multiplying the weights through again and stacking together $\mathbf{D}_i$ for each correspondence into $\mathbf{D}$ (real $2n$ x 4 matrix), we can arrive at the following constrained least squares objective:

$$||\mathbf{D}\mathbf{q}||^2 = \mathbf{q}^T\mathbf{D}^T\mathbf{D}\mathbf{q} = \mathbf{q}^T\Big(\sum_i w_i'\mathbf{D}_i^T\mathbf{D}_i\Big)\mathbf{q} = \mathbf{q}^T\mathbf{G}_P\mathbf{q}$$

$$\min_{\mathbf{q}} \mathbf{q}^T\mathbf{G}_P\mathbf{q}, \ s.t. \ ||\mathbf{q}|| = 1 \tag{12}$$

The formulated objective in Eq. (12) is equivalent to the original problem statement, and the solution is well known as the eigenvector corresponding to the smallest eigenvalue of $\mathbf{G}_P$. Using Eq. (35) again, we can map $\mathbf{q}$ back to a special unitary matrix $\mathbf{U}$ giving a solution to the problem. Note that $-\mathbf{q}$ is also a solution since eigenvectors are only unique up to scale. However, the sign is irrelevant as $\mathbf{q}$ and $-\mathbf{q}$ map to the same rotation due to the double cover of quaternions over $SO(3)$ in Eq. (36). For further theoretical details on this solution, see Appendix C.

## 2.2 Approximation via Möbius Transformations

We can approximate the previous solution in the complex domain by first estimating an optimal Möbius transformation $\mathbf{M}$ and mapping it to a special unitary matrix. Relaxing the special unitary

conditions in Eq. (9), we can treat $\mathbf{u}$ as a flattened form of $\mathbf{M}$, leading to a modified constraint $\mathbf{A}'_i$ that holds when $\mathbf{M}$ aligns a stereographic point pair:

$$\mathbf{m} = vec(\mathbf{M}) = \begin{bmatrix} \sigma & \xi & \gamma & \delta \end{bmatrix}^T$$

$$\begin{bmatrix} -z_i & -1 & p_i z_i & p_i \end{bmatrix} \mathbf{m} = \mathbf{A}'_i \mathbf{m} = 0 \tag{13}$$

Note that Eq. (13) does not preserve the metric in Eq. (4) between $p_i$ and transformed point $\mathbf{\Phi_M}(z_i)$. We can stack each $\mathbf{A}'_i$ into matrix $\mathbf{A}'$ ($n$ x 4 complex matrix) and similarly estimate the best (in a least squares sense) Möbius transformation aligning the points as:

$$\mathbf{G}_M = \mathbf{A}'^H \mathbf{A}' = \sum_i \mathbf{A}'^H_i \mathbf{A}'_i$$

$$\min_{\mathbf{m}} \mathbf{m}^H \mathbf{G}_M \mathbf{m} \ \ s.t. \ \ ||\mathbf{m}|| = 1 \tag{14}$$

The constraint in Eq. (14) is necessary to prevent trivial solutions, but the choice of quadratic constraint on $\mathbf{m}$ is arbitrary. With our constraint choice, the optimal $\mathbf{m}$ is the complex eigenvector corresponding to the smallest eigenvalue of $\mathbf{G}_M$. Since $\mathbf{G}_M$ is positive semidefinite and Hermitian ($\mathbf{G}_M^H = \mathbf{G}_M$) by construction, the eigenvalues are real and nonnegative, facilitating straightforward ordering. If $n < 4$, $\mathbf{m}$ can be obtained directly from the kernel of $\mathbf{A}'$. Either way, the solution is not unique as eigenvectors and kernel vectors can be scaled arbitrarily, particulary by a phase $e^{i\theta}$. However, by Eq. (42), scaled Möbius transformations are equivalent, so our result properly defines the transformation.

Given $\mathbf{m}$, we can reshape it into $\mathbf{M}$ and scale $\mathbf{M}$ to $\mathbf{M}^* = \det(\mathbf{M})^{-\frac{1}{2}} \mathbf{M}$ (allowed since the scale of $\mathbf{M}$ is arbitrary) so that $\det(\mathbf{M}^*) = 1$. It is known that the closest unitary matrix to $\mathbf{M}^*$ in the Frobenius sense can be computed by $\mathbf{U}\mathbf{V}^H$ (Keller, 1975), where $\mathbf{U}$ and $\mathbf{V}^H$ are from the singular value decomposition $\mathbf{M}^* = \mathbf{U}\mathbf{\Sigma}\mathbf{V}^H$. Since $\det(\mathbf{M}^*) = 1$, the nearest unitary matrix to $\mathbf{M}^*$ is special unitary (proof in Appendix B.3.1) and is in fact the approximate solution. Note that this matrix is not necessarily the nearest special unitary matrix to $\mathbf{M}$ itself. By normalizing the determinant, we prevent the rotation mapping from being affected by arbitrary phase scalings of $\mathbf{m}$.

### 2.3 3D Sphere Solution

If our inputs are given as unit observations in 3D, we could project them by $\psi$ and use the earlier solution. However, through Eqs. (37) and (38), we see that we can act directly on 3D vectors with special unitary matrices which suggests an alternative formulation. Upon examining the structure of the matrices that $\chi$ maps to, one can show that Eq. (1) can be equivalently expressed as:

$$\chi(\mathbf{a}_i) \mapsto \mathbf{Z}_i, \ \chi(\mathbf{b}_i) \mapsto \mathbf{P}_i$$

$$\sum_i w_i ||\mathbf{b}_i - \mathbf{R}\mathbf{a}_i||^2 = \frac{1}{2} \sum_i w_i ||\mathbf{P}_i - \mathbf{U}\mathbf{Z}_i\mathbf{U}^H||_F^2 \tag{15}$$

where $|| \cdot ||_F$ denotes the Frobenius norm and $\mathbf{U}$ is the special unitary matrix that maps to $\mathbf{R}$. The Frobenius norm is unitarily invariant, so we may multiply the inside expression on the right by $\mathbf{U}$ to obtain a new target objective and corresponding constraint:

$$\frac{1}{2} \sum_i w_i ||\mathbf{P}_i\mathbf{U} - \mathbf{U}\mathbf{Z}_i||_F^2 = 0 \implies \sqrt{\frac{w_i}{2}}(\mathbf{P}_i\mathbf{U} - \mathbf{U}\mathbf{Z}_i) = 0 \tag{16}$$

We arrive at a linear constraint again via special unitary matrices. Inspecting the matrix within the Frobenius norm reveals that the loss contribution from the top row elements is identical to that of the bottom row elements. Consequently, we only need to compute the loss from a single row, allowing us to eliminate the factor of $\frac{1}{2}$ from equation Eq. (16). With $\mathbf{a}_i = (x_i, y_i, z_i)$ and $\mathbf{b}_i = (m_i, n_i, p_i)$, we can write the following complex constraint:

$$\sqrt{w_i} \begin{bmatrix} (m_i - x_i)i & y_i - z_i i & 0 & -n_i - p_i i \\ -y_i - z_i i & (x_i + m_i)i & n_i + p_i i & 0 \end{bmatrix} \mathbf{u} = \sqrt{w_i}\mathbf{C}_i\mathbf{u} = 0 \tag{17}$$

$\mathbf{C}_i$ has a rank of at most 1 if $\mathbf{a}$ and $\mathbf{b}$ have the same magnitude. We reformulate the constraint, once again breaking the complex terms of $\mathbf{u}$ into their real components. This yields the following linear

constraint in terms of quaternion parameters:

$$\sqrt{w_i} \begin{bmatrix} 0 & x_i - m_i & y_i - n_i & z_i - p_i \\ m_i - x_i & 0 & -z_i - p_i & y_i + n_i \\ n_i - y_i & z_i + p_i & 0 & -x_i - m_i \\ p_i - z_i & -y_i - n_i & x_i + m_i & 0 \end{bmatrix} \mathbf{q} = \sqrt{w_i} \mathbf{Q}_i \mathbf{q} = 0 \tag{18}$$

Note that $\mathbf{Q}_i$ is a 4x4 skew-symmetric matrix and has at most rank 2 if $\mathbf{a}$ and $\mathbf{b}$ have the same magnitude. As a result, our optimization now becomes:

$$\sum_i w_i \mathbf{Q}_i^T \mathbf{Q}_i = -\sum_i w_i \mathbf{Q}_i^2 = \mathbf{G}_S$$

$$\min_{\mathbf{q}} \mathbf{q}^T \mathbf{G}_S \mathbf{q} \ \ s.t. \ ||\mathbf{q}|| = 1 \tag{19}$$

The solution is once again the eigenvector corresponding to the smallest eigenvalue of $\mathbf{G}_S$.

## 3 Optimization Methods From Linear Quaternion Constraints

Our previous general solutions are notably distinct from other methods as they allow for the principled construction of linear constraints (Eqs. (11) and (18)) on quaternion parameters. We discuss some applications and desirable properties of these results.

### 3.1 Residual Based Optimization

While Wahba's problem admits a direct solution, many related rotation estimation tasks require iterative methods. These often involve repeatedly evaluating per-observation losses for a candidate quaternion. Examples include alternative loss functions like the absolute chordal metric ($L_1$ distance) or robust approaches such as iteratively reweighted least squares (IRLS). In these settings, our linear constraints serve as a drop-in, efficient method for residual computation. The stereographic formulation in Eq. (11) is especially appealing as it is far more compact (8 elements versus 12 for Eq. (18)) while avoiding branching in construction, especially in the general case of Appendix C.2.

### 3.2 Constrained Optimization

When the constraints for an observation pair hold exactly, our formulas yield a convenient analytical characterization of all rotations that align the pair. A practical use case for this is rotation estimation with an axis prior (e.g. a gravity vector measurement from an IMU). Traditional methods rely on sequential rotations or intermediate coordinate frames to simplify the problem (Magner and Zee, 2018; Chandrasekhar, 2024). In contrast, because both Eqs. (11) and (18) reduce to rank 2 in this setting, we can linearly express two quaternion parameters in terms of the other two and solve directly and efficiently in a reduced space, eliminating the need for intermediate frames.

### 3.3 Two-Point Case for Wahba's Problem

More generally speaking, when the constraints hold exactly for one or more observation pairs (i.e. noiseless scenarios), we can obtain the solution from the kernel of those constraints in closed-form. For example, with two noiseless 3D sphere observation pairs, the aligning rotation can be given by:

$$\tilde{\mathbf{q}} = \begin{bmatrix} (\mathbf{a}_1 + \mathbf{b}_1) \cdot (\mathbf{a}_2 - \mathbf{b}_2) \\ (\mathbf{a}_1 - \mathbf{b}_1) \times (\mathbf{a}_2 - \mathbf{b}_2) \end{bmatrix} \tag{20}$$

where $\tilde{\mathbf{q}}$ denotes the unnormalized form of rotation $\mathbf{q}$. Appendix D describes our methods to robustly and efficiently construct these rotations of exact alignment. These simple kernel formulations are key to enabling our solutions to the case of Wahba's problem when $n = 2$.

**Weighted** Wahba's problem for the two-point case is well known to have closed-form expressions (Shuster and Oh, 1981; Mortari, 1997; Markley, 2002). We propose an alternate solution which is given by the weighted average of the two (unnormalized) rotations that each noiselessly align the cross products of the reference and target sets, along with one of the two corresponding

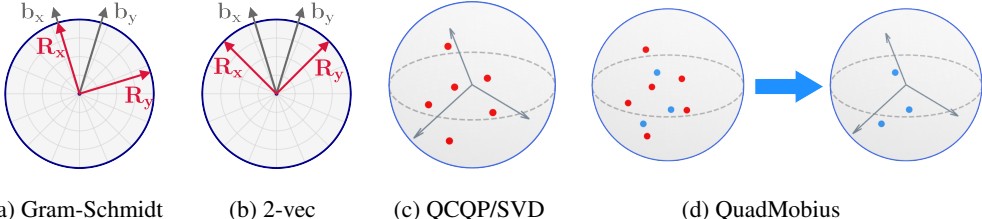

(a) Gram-Schmidt      (b) 2-vec      (c) QCQP/SVD      (d) QuadMobius

Figure 1: (a)-(b) Illustration of difference between Gram-Schmidt and 2-vec in 2D. $\mathbf{b_x}$, $\mathbf{b_y}$ are predicted axes directions from the model, and $\mathbf{R_x}$, $\mathbf{R_y}$ are the orthogonalized coordinate axes from each mapping. Gram-Schmidt favors $\mathbf{b_x}$, aligning $\mathbf{R_x}$ with it greedily while 2-vec uses $\mathbf{b_x}, \mathbf{b_y}$ in a balanced way. (c)-(d) Conceptual illustration of QCQP, SVD, and QuadMobius maps in context of Wahba's problem in 3D. QCQP/SVD can be interpreted as direct projection of target points (red) to an orthogonal frame. QuadMobius first maps those points to an intermediate representation—a Möbius transformation, defined by three points (blue)—before projecting to an $SU(2)$ rotation.

observation pairs (proof in Appendix B.4.1). Using the average rotation definition from Markley et al. (2007) (i.e. in Frobenius sense for $SO(3)$), the solution is:

$$\mathbf{n}_1 = \mathbf{a}_1 \times \mathbf{a}_2, \quad \mathbf{n}_2 = \sqrt{\frac{||\mathbf{a}_1 \times \mathbf{a}_2||^2}{||\mathbf{b}_1 \times \mathbf{b}_2||^2}}(\mathbf{b}_1 \times \mathbf{b}_2), \quad \tilde{\mathbf{q}}_i = \begin{bmatrix} (\mathbf{a}_i + \mathbf{b}_i) \cdot (\mathbf{n}_1 - \mathbf{n}_2) \\ (\mathbf{a}_i - \mathbf{b}_i) \times (\mathbf{n}_1 - \mathbf{n}_2) \end{bmatrix}$$

$$\tau = (w_1 - w_2)||\tilde{\mathbf{q}}_1||^2||\tilde{\mathbf{q}}_2||^2, \quad \omega = 2w_1||\tilde{\mathbf{q}}_2||^2(\tilde{\mathbf{q}}_1 \cdot \tilde{\mathbf{q}}_2)$$

$$\nu = 2w_2||\tilde{\mathbf{q}}_1||^2(\tilde{\mathbf{q}}_1 \cdot \tilde{\mathbf{q}}_2), \quad \mu = \tau + \sqrt{\tau^2 + \omega\nu}$$

$$\mathbf{q} = \frac{\mu\tilde{\mathbf{q}}_1 + \nu\tilde{\mathbf{q}}_2}{\sqrt{||\tilde{\mathbf{q}}_1||^2\mu^2 + ||\tilde{\mathbf{q}}_2||^2\nu^2 + 2(\tilde{\mathbf{q}}_1 \cdot \tilde{\mathbf{q}}_2)\mu\nu}} \tag{21}$$

where $\tilde{\mathbf{q}}_1 \cdot \tilde{\mathbf{q}}_2$ denotes the usual vector dot product between $\tilde{\mathbf{q}}_1$ and $\tilde{\mathbf{q}}_2$. See Appendix B.4.3 for derivation and additional details.

**Unweighted** In the case of $w_1 = w_2$, the optimal rotation simplifies to the rotation which exactly aligns $\mathbf{a}_1 + \mathbf{a}_2$ to $\mathbf{b}_1 + \mathbf{b}_2$ and $\mathbf{a}_1 - \mathbf{a}_2$ to $\mathbf{b}_1 - \mathbf{b}_2$ (proof in Appendix B.4.2). This is given by:

$$\mathbf{s}_1 = \mathbf{a}_1 + \mathbf{a}_2, \quad \mathbf{s}_2 = \sqrt{\frac{1 + \mathbf{a}_1 \cdot \mathbf{a}_2}{1 + \mathbf{b}_1 \cdot \mathbf{b}_2}}(\mathbf{b}_1 + \mathbf{b}_2)$$

$$\mathbf{d}_1 = \mathbf{a}_1 - \mathbf{a}_2, \quad \mathbf{d}_2 = \sqrt{\frac{1 - \mathbf{a}_1 \cdot \mathbf{a}_2}{1 - \mathbf{b}_1 \cdot \mathbf{b}_2}}(\mathbf{b}_1 - \mathbf{b}_2)$$

$$\tilde{\mathbf{q}} = \begin{bmatrix} (\mathbf{s}_1 + \mathbf{s}_2) \cdot (\mathbf{d}_1 - \mathbf{d}_2) \\ (\mathbf{s}_1 - \mathbf{s}_2) \times (\mathbf{d}_1 - \mathbf{d}_2) \end{bmatrix} \tag{22}$$

The aligning rotation formulas are given in the form of equation Eq. (20) for simplicity, but in practice we use the approach described in Appendix D.2 for robustness. In that case, singular cases only arise when $\mathbf{a}_1 \times \mathbf{a}_2 = 0$ or $\mathbf{b}_1 \times \mathbf{b}_2 = 0$ where no unique solution exists, and a particular one may be obtained via the special unitary constraints in equation Eq. (17) (see Appendix B.4.4). Notably, the two solutions above are optimal in the sense of Wahba's problem and simplified compared to existing two-point methods, especially for the unweighted case (see Table 2).

An example use case of these methods is estimating the orientation of a camera given an image of a rectangle. Under a pinhole camera model, the image of a 3D rectangle adheres to the rules of perspective geometry. Since the rectangle's opposite edges are parallel in 3D, their projections in the image converge at vanishing points that represent the direction of these lines in the camera's frame. Because the two sets of parallel edges in the rectangle are orthogonal in 3D, the corresponding vanishing points should also be orthogonal. However, due to measurement noise, this orthogonality is often violated. Our two point solutions can recover the best estimate of the camera's orientation in these cases.

## 4 REPRESENTATIONS FOR LEARNING ROTATIONS

Based on previous formulations, we introduce two higher-dimensional representations for learning rotations. See Appendix B.2 for derivation details and Appendix F for further theoretical support and explanation of both representations.

**2-vec** The first is based on our formula for the optimal rotation from two unweighted observations and is denoted 2-vec. Similar to the Gram-Schmidt map in Zhou et al. (2019), 2-vec interprets a 6D output vector from a model as target 3D $x$ and $y$ axes (denoted $\mathbf{b}_x$, $\mathbf{b}_y$). Unlike the Gram-Schmidt method which greedily orthogonalizes the two vectors by assuming the x-axis prediction is correct, 2-vec maps the two vectors to a rotation optimally in the sense of Wahba's problem, balancing error from both axis predictions (Fig. 4). Eq. (22) could be used, but since the reference points are the $x, y$ coordinate axes, we can instead obtain a rotation matrix in a simpler fashion through the same principle:

$$\mathbf{b}'_y = \frac{||\mathbf{b}_x||}{||\mathbf{b}_y||}\mathbf{b}_y, \quad \mathbf{b}^+ = \frac{\mathbf{b}_x + \mathbf{b}'_y}{||\mathbf{b}_x + \mathbf{b}'_y||}, \quad \mathbf{b}^- = \frac{\mathbf{b}_x - \mathbf{b}'_y}{||\mathbf{b}_x - \mathbf{b}'_y||}$$

$$\mathbf{R} = \left[ \tfrac{1}{\sqrt{2}}(\mathbf{b}^+ + \mathbf{b}^-), \ \tfrac{1}{\sqrt{2}}(\mathbf{b}^+ - \mathbf{b}^-), \ \mathbf{b}^- \times \mathbf{b}^+ \right] \in SO(3) \tag{23}$$

This method has a similar singular region and computational complexity as that of Gram-Schmidt.

**QuadMobius** A second parameterization is based on the approximation from Section 2.2 involving Möbius transformations. Taking inspiration from the approach in Peretroukhin et al. (2020), a (real) 16D network output $\Theta = \{\theta_i : i = 1 \ldots 16\}$ is arranged into the unique complex elements of $\mathbf{G}_M$ as below:

$$\mathbf{G}_M(\Theta) = \begin{bmatrix} \theta_1 & \theta_2 + \theta_3 i & \theta_4 + \theta_5 i & \theta_6 + \theta_7 i \\ \theta_2 - \theta_3 i & \theta_8 & \theta_9 + \theta_{10} i & \theta_{11} + \theta_{12} i \\ \theta_4 - \theta_5 i & \theta_9 - \theta_{10} i & \theta_{13} & \theta_{14} + \theta_{15} i \\ \theta_6 - \theta_7 i & \theta_{11} - \theta_{12} i & \theta_{14} - \theta_{15} i & \theta_{16} \end{bmatrix} \tag{24}$$

$\mathbf{G}_M(\Theta)$ is Hermitian with real (and assumed distinct) eigenvalues where we can select the eigenvector $\mathbf{m}$ corresponding to its smallest eigenvalue. After reshaping $\mathbf{m}$ to a Möbius transformation $\mathbf{M}$, we can map to a rotation by the approximation procedure in Section 2.2. The procedure can be performed via singular value decomposition ($\mathbf{M} = \mathbf{U}\boldsymbol{\Sigma}\mathbf{V}^H$) to obtain a special unitary matrix $\mathbf{Q}$:

$$\mathbf{Q} = \overline{\sqrt{det(\mathbf{U}\mathbf{V}^H)}}\mathbf{U}\mathbf{V}^H \in SU(2) \tag{25}$$

Alternatively, we can algebraically solve for $\mathbf{Q}$ as follows:

$$\mathbf{M}^* = \sqrt{\frac{\overline{det(\mathbf{M})}}{|det(\mathbf{M})|(2|det(\mathbf{M})| + Tr(\mathbf{M}^H\mathbf{M}))}}\mathbf{M}$$

$$\mathbf{Q} = \mathbf{M}^* + adj(\mathbf{M}^*)^H \in SU(2) \tag{26}$$

where $Tr(\cdot)$ denotes the trace and $adj(\cdot)$ denotes the adjugate. In both cases, $\mathbf{Q}$ is mapped to a quaternion via Eqs. (35) and (45), and $\mathbf{M}$ is assumed to be nonsingular. We denote the SVD method **QuadMobiusSVD** and the algebraic method **QuadMobiusAlg**. With these maps and our assumptions (observed valid in practice), we define a full mapping from $\Theta$ to $\mathbf{q}$ that has a defined numerical derivative for backpropagation (see Appendix E for derivative formulas).

Intuitively, QuadMobius derives its strength from its two-stage structure illustrated in Fig. 1d. Whereas single-step maps often face an inherent trade-off between providing informative gradients and remaining robust to noisy inputs, QuadMobius decouples these roles. The eigendecomposition stage learns a stable latent space (identified with the least-squares estimate of a Möbius transformation) that's buffered against input noise and arbitrary scalings, while the subsequent projection to $SU(2)$ provides a strong, structured pathway for learning. Our experiments in Appendix F reinforce this interpretation. Moreover, we remark that since QuadMobius combines ideas from Levinson et al. (2020) and Peretroukhin et al. (2020), it inherits several of their theoretical properties including interpretation as a Bingham belief (Kent, 1994) and differentiability (Magnus, 1985; Wan and Zhang, 2019).

## 5 EXPERIMENTS

### 5.1 WAHBA'S PROBLEM

Synthetic experiments are performed to validate the proposed methods for Wahba's problem. For each trial, a ground truth quaternion rotation $\mathbf{q}_{gt}$ is randomly sampled from $S^3$, and $n$ reference points are randomly sampled from $S^2$. The reference points are rotated by $\mathbf{q}_{gt}$ to obtain target observations. Gaussian noise is added to each component of each target observation, and the target observations are subsequently re-normalized afterward. Weights are randomly sampled between 0 and 1. Accuracy is measured by the angular distance $\theta_{err} = cos^{-1}(2(\mathbf{q}_{est} \cdot \mathbf{q}_{gt})^2 - 1)$ in degrees between the estimated rotation $\mathbf{q}_{est}$ and $\mathbf{q}_{gt}$, where $(\cdot, \cdot)$ denotes the usual vector dot product. Numerical results shown in Appendix.

We first test our solutions to Wahba's problem for both 3D and stereographic inputs (Eqs. (12) and (19)). The input for the latter is created by projecting the 3D points by $\psi$. We also test the approximate solution in Section 2.2. The solutions to all three are obtained by eigendecomposition using Jacobi's eigenvalue algorithm. For validation, we compare against several quaternion solvers introduced over the past decades. For the two-point case, we also compare against the closed-form solutions in Markley (2002) and Shuster and Oh (1981). All solutions were reimplemented and optimized similarly in C++17 and compiled with the flag -O3. We perform one million trials for each configuration.

| Algorithm | $n = 3$ | | | $n = 100$ | | |
| | $\epsilon = 1e^{-5}$ | $\epsilon = 0.1$ | Timings | $\epsilon = 1e^{-5}$ | $\epsilon = 0.1$ | Timings |
|---|---|---|---|---|---|---|
| Q-method Davenport (1968) | 7.4676e-4 | 7.4868 | 3.583 | 1.2487e-4 | 1.2551 | 5.375 |
| QUEST Shuster and Oh (1981) | 7.4676e-4 | 7.4868 | 0.250 | 1.2487e-4 | 1.2551 | 1.875 |
| ESOQ2 Mortari (1997) | 7.4694e-4 | 7.4869 | 0.375 | 1.2487e-4 | 1.2551 | 2.000 |
| FLAE Wu et al. (2018) | 7.4676e-4 | 7.4868 | 0.333 | 1.2487e-4 | 1.2551 | 1.875 |
| OLAE Mortari et al. (2007) | 7.7118e-4 | 7.8639 | 0.208 | 1.3120e-4 | 1.5952 | 2.167 |
| Ours ($\mathbf{G}_P$, Eq. (12)) | 7.4676e-4 | 7.4868 | 4.084 | 1.2487e-4 | 1.2551 | 9.917 |
| Ours ($\mathbf{G}_S$, Eq. (19)) | 7.4676e-4 | 7.4868 | 3.625 | 1.2487e-4 | 1.2551 | 6.500 |
| Ours ($\mathbf{G}_M$, Eq. (14)) | 1.2614e-3 | 12.608 | 0.917 | 3.5870e-4 | 3.7782 | 41.875 |

Table 1: Results of various Wahba's Problem solvers against varying noise levels with $n = \{3, 100\}$. Accuracy values reported are median $\theta_{err}$, and timing values are median runtimes in microseconds. Timings taken with $\epsilon_{noise}$=0.1. See Section 5.1 for more info.

Table 1 confirms that our optimal solvers match the accuracy of Davenport's Q-method in the general case. In contrast, our Möbius approximation demonstrates a sensitivity to noise and could likely benefit from a normalization step common in real homography estimation (Hartley and Zisserman, 2004). However, when paired with a stable, learned input representation (as shown in later experiments), the method's sensitivity becomes an asset by providing strong gradients for learning.

| Algorithm | x | $\div$ | $\sqrt{}$ | $5^{th}$ | $50^{th}$ | $95^{th}$ |
|---|---|---|---|---|---|---|
| QUEST (Shuster and Oh, 1981) | 89 / 99 | **1 / 1** | **3 / 3** | 3.3082 / 3.4115 | 9.1727 / 9.3970 | 27.0520 / 27.1371 |
| Fast 2 Vec (Markley, 2002) | 72 / 78 | 3 / 3 | 4 / 4 | 3.3082 / 3.4115 | 9.1727 / 9.3970 | 27.0520 / 27.1371 |
| SUPER (Ours) | **29 / 74** | 3 / 2 | **3 / 3** | 3.3082 / 3.4115 | 9.1727 / 9.3970 | 27.0520 / 27.1371 |

Table 2: Operation counts and $\theta_{err}$ percentiles ($\epsilon_{noise} = 0.1$) for two-point Wahba's problem solvers. Values given for unweighted/weighted algorithms without edge case handling. Bold indicates best.

Table 2 similarly confirms that our two-point methods achieve the same optimal results as existing solvers. By utilizing unnormalized rotations, our weighted algorithm minimizes normalization costs, streamlining the compute. Most notably, in the unweighted case, our tailored solution only requires roughly a third of the multiplications of other methods, marking a significant gain in efficiency.

|  | Chair | | | | Sofa | | | | Toilet | | | |
|---|---|---|---|---|---|---|---|---|---|---|---|---|
|  | Mean | Med. | $\text{Acc}_5$ | $\text{Acc}_{10}$ | Mean | Med. | $\text{Acc}_5$ | $\text{Acc}_{10}$ | Mean | Med. | $\text{Acc}_5$ | $\text{Acc}_{10}$ |
| Euler | 21.479 | 10.777 | 0.129 | 0.457 | 22.033 | 9.462 | 0.153 | 0.529 | 14.495 | 8.375 | 0.197 | 0.604 |
| Quat | 23.640 | 12.664 | 0.083 | 0.350 | 23.426 | 10.778 | 0.128 | 0.452 | 14.959 | 9.913 | 0.128 | 0.511 |
| GS | 13.606 | 6.320 | 0.350 | 0.738 | 15.015 | 5.469 | 0.441 | **0.801** | 6.586 | 3.708 | 0.682 | 0.915 |
| QCQP | 13.131 | 5.786 | 0.416 | 0.773 | 13.916 | 5.476 | 0.436 | 0.795 | 6.070 | **3.452** | **0.730** | 0.929 |
| SVD | 13.061 | 5.815 | 0.412 | 0.773 | 14.967 | 5.812 | 0.406 | 0.774 | 6.135 | 3.502 | 0.710 | **0.930** |
| 2-vec | **12.544** | 6.100 | 0.380 | 0.751 | 15.077 | 6.217 | 0.364 | 0.753 | 6.069 | 3.483 | 0.713 | 0.926 |
| QMAlg | 12.604 | **5.696** | **0.425** | **0.783** | 14.336 | 5.657 | 0.419 | 0.793 | 6.079 | 3.590 | 0.714 | **0.930** |
| QMSVD | 13.157 | 6.211 | 0.366 | 0.748 | **13.683** | **5.421** | **0.443** | 0.799 | **6.026** | 3.601 | 0.699 | 0.926 |

Table 3: $\theta_{err}$ mean/median and accuracy (subscript indicates threshold) on 3D shape alignment for different ModelNet10-SO3 categories (Liao et al., 2019). Bold indicates best, underline indicates second best.

## 5.2 LEARNING EXPERIMENTS

We conduct several experiments to evaluate our proposed rotation representations. The primary loss function is the squared Frobenius norm $||\mathbf{R}_{pred} - \mathbf{R}_{gt}||^2_F$, which we refer to as **Chordal L2**, where $\mathbf{R}_{pred}$ is the predicted rotation and $\mathbf{R}_{gt}$ is the ground truth. For quaternion outputs, Chordal L2 is computed same as Peretroukhin et al. (2020). We compare our representations—**2-vec**, QuadMobiusAlg (**QMAlg**), and QuadMobiusSVD (**QMSVD**)—against several baselines: **Euler** angles (Tait-Bryan YXZ), **Quat** (quaternion), **GS** (Gram-Schmidt) (Zhou et al., 2019), **QCQP** (Peretroukhin et al., 2020), and **SVD** (Levinson et al., 2020). In both QuadMobius variants, we use the algebraic method in the forward pass to avoid SVD computation and isolate differences to the backward pass. This section presents results on three public benchmarks. Additional synthetic experiments exploring different learning conditions are included in Appendix G.2.1, and full training details are provided in Appendix G.1.

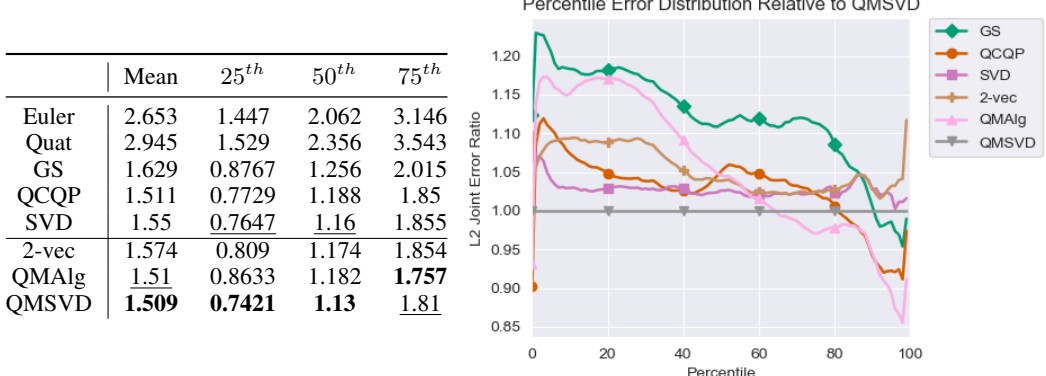

|  | Mean | $25^{th}$ | $50^{th}$ | $75^{th}$ |
|---|---|---|---|---|
| Euler | 2.653 | 1.447 | 2.062 | 3.146 |
| Quat | 2.945 | 1.529 | 2.356 | 3.543 |
| GS | 1.629 | 0.8767 | 1.256 | 2.015 |
| QCQP | 1.511 | 0.7729 | 1.188 | 1.85 |
| SVD | 1.55 | 0.7647 | 1.16 | 1.855 |
| 2-vec | 1.574 | 0.809 | 1.174 | 1.854 |
| QMAlg | 1.51 | 0.8633 | 1.182 | **1.757** |
| QMSVD | **1.509** | **0.7421** | **1.13** | 1.81 |

Figure 2: Results of implicit learning for Inverse Kinematics task (Zhou et al., 2019). Left: Mean and percentile L2 distance error (cm) of predicted joint locations. Bold indicates best, underline indicates second best. Right: Ratios of joint errors relative to QMSVD across error percentiles (Euler/Quat omitted due to large ratios).

**ModelNet10-SO3** We first evaluate the representations on the 3D shape alignment task from Liao et al. (2019) using the ModelNet10-SO3 dataset. This dataset comprises of images of 3D CAD models under uniformly sampled rotations with multiple object models per category. The task is to predict the object's orientation directly from its image. Table 3 reports the results on three object categories, chosen for their low rotational symmetry following the choice in Levinson et al. (2020).

**Inverse Kinematics** Next, we test the representations on an implicit learning task, applying them to the inverse kinematics task from Zhou et al. (2019). Given 3D human pose joint locations (from

real-world motion capture data), a network predicts the joint orientations relative to a reference pose and uses a fixed forward kinematics function to obtain predicted joint locations. The distance loss is applied between the predicted and given joint locations. In this task, the rotations are used as implicit representations through which the gradients must flow rather than direct prediction targets. Fig. 2 compares the results of the different learning representations on this task.

**Camera Pose Estimation**   Finally, we replicate the experiment from Walch et al. (2017) which utilizes an LSTM to directly regress a camera's pose from real world images. Training requires simultaneously optimizing over both the camera's orientation and translation. Data comes from the Cambridge Landmarks dataset (Kendall et al., 2015) which includes labels estimated from traditional structure from motion pipelines. The results are seen in Table 4 from training on select scenes, following the choice of Chen et al. (2022).

**Results**   Overall, the proposed representations demonstrated strong performance and versatility across the three benchmark tasks. Despite its lower dimensionality, 2-vec proved competitive, occasionally achieving the best result. Notably, it typically outperforms Gram-Schmidt, positioning itself as an attractive alternative for same compute and dimensionality. The QuadMobius approaches showed their potential by achieving the top result in nearly all experiments over favorites like SVD and QCQP.

| | King's College | | | | Shop Facade | | | | Old Hospital | | | |
|---|---|---|---|---|---|---|---|---|---|---|---|---|
| | Mean | $25^{th}$ | $50^{th}$ | $75^{th}$ | Mean | $25^{th}$ | $50^{th}$ | $75^{th}$ | Mean | $25^{th}$ | $50^{th}$ | $75^{th}$ |
| Euler | 4.192 | 2.403 | 3.684 | 5.509 | 6.826 | 4.129 | 6.050 | 9.305 | 4.748 | 2.204 | 3.247 | 6.162 |
| Quat | 2.759 | 1.367 | 2.251 | 3.499 | 6.604 | **3.762** | 5.339 | 8.153 | 4.570 | 2.486 | 3.377 | 5.546 |
| GS | 3.298 | 1.764 | 2.583 | 4.137 | 6.559 | 4.376 | 5.660 | 8.343 | 4.295 | **1.897** | 3.070 | 5.698 |
| QCQP | 3.204 | 1.540 | 2.537 | 4.129 | 6.802 | 3.901 | 5.797 | 8.539 | 4.454 | 2.156 | 3.304 | 6.267 |
| SVD | 3.292 | 1.589 | 2.624 | 4.110 | 7.117 | 4.157 | 5.647 | 8.370 | 4.574 | 2.420 | 3.485 | 5.961 |
| 2-vec | 3.085 | 1.536 | 2.371 | 4.014 | 7.118 | 3.789 | 5.762 | 8.957 | **4.294** | 2.085 | **2.950** | **5.292** |
| QMAlg | **2.631** | **1.337** | **2.052** | **3.267** | **6.317** | 4.050 | **5.268** | **7.758** | 4.426 | 2.035 | 3.238 | 5.640 |
| QMSVD | 2.706 | 1.391 | 2.177 | 3.345 | 6.715 | 4.074 | 5.710 | 8.947 | 4.409 | 2.077 | 3.146 | 5.744 |

Table 4: Mean and percentile $\theta_{err}$ of predicted rotations from direct pose prediction on different scenes in Cambridge Landmarks Dataset (Kendall et al., 2015). Bold indicates best, underline indicates second best.

## 6   CONCLUSION

This paper demonstrated the utility of special unitary matrices for rotation estimation. Several new formulas and algorithms were presented from this perspective for the real and complex domains, tackling Wahba's problem and rotation representations in neural networks. Various experiments confirmed the potential of these approaches. Future work may include further solidifying the theoretical and empirical foundations of our rotation representations and applying special unitary matrices to other tasks such as analytical camera pose estimation.

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

# Special Unitary Parameterized Estimators of Rotation

## Appendix

## A    MATHEMATICAL BACKGROUND AND DEFINITIONS

The mathematical background for special unitary matrices and related concepts is briefly reviewed. The formulas are all established and generally known. A complex square matrix $\mathbf{U}$ is defined as unitary if:

$$\mathbf{U}\mathbf{U}^H = \mathbf{U}^H\mathbf{U} = \mathbf{I}, \;\; |det(\mathbf{U})| = 1 \tag{27}$$

where $^H$ denotes the conjugate transpose, $|\cdot|$ denotes complex magnitude, and $det(\cdot)$ denotes determinant. The matrix is *special unitary* if it has the additional restriction that $det(\mathbf{U}) = 1$ exactly.

Stereographic projection $\psi$ is an invertible mapping of the sphere $S^2 = \{(x_s, y_s, z_s) \mid x_s^2 + y_s^2 + z_s^2 = 1\}$ from the point $\mathbf{p}^* = (0, 0, -1)$ to the complex plane and is given by:

$$\psi_{\mathbb{C}}(\mathbf{a}) : \; \frac{x_s}{1 + z_s} + \frac{y_s}{1 + z_s}i = x_p + y_p i = z \tag{28}$$

$$\psi_{\mathbb{C}}^{-1}(z) : \; \left( \frac{2x_p}{1 + x_p^2 + y_p^2}, \frac{2y_p}{1 + x_p^2 + y_p^2}, \frac{1 - x_p^2 - y_p^2}{1 + x_p^2 + y_p^2} \right) \tag{29}$$

where $\mathbf{a} \in S^2$ and $z \in \mathbb{C}$. This projection is visualized in Fig. 3. Note that $\psi_{\mathbb{C}}$ is undefined when $\mathbf{a} = \mathbf{p}^*$. To overcome this, the map is extended to the complex projective space $\mathbb{CP}^1$ which includes the point at infinity so we can define $\psi_{\mathbb{CP}}(\mathbf{p}^*) = \infty$. The projection is now redefined below with equivalence relations:

$$\psi_{\mathbb{CP}}(\mathbf{a}) \mapsto \begin{cases} \begin{bmatrix} z \\ 1 \end{bmatrix} \sim \lambda \begin{bmatrix} z \\ 1 \end{bmatrix}, & \mathbf{a} \neq \mathbf{p}^* \\ \infty \sim \begin{bmatrix} \lambda \\ 0 \end{bmatrix}, & \mathbf{a} = \mathbf{p}^* \end{cases} \tag{30}$$

$$\lambda \in \mathbb{C}, \; \lambda \neq 0, \; \psi_{\mathbb{CP}}^{-1}(\psi_{\mathbb{CP}}(\mathbf{a})) = \mathbf{a}$$

In this paper, our use of $\psi$ generally refers to $\psi_{\mathbb{CP}}$. From the above definition, $\psi(\mathbf{a})$ can be arbitrarily scaled, and $\psi$ bijectively maps the entire sphere to the complex projective space. Note that this mapping is not unique, particularly since choice of $\mathbf{p}^*$ is arbitrary (any point on $S^2$ is valid). We will use the specific projection defined above for this paper as it is convenient for image processing.

A special unitary matrix $\mathbf{U} \in SU(2)$ can generally be written as:

$$\mathbf{U} = \begin{bmatrix} \alpha & \beta \\ -\bar{\beta} & \bar{\alpha} \end{bmatrix} \tag{31}$$

$$\alpha\bar{\alpha} + \beta\bar{\beta} = 1, \;\; \alpha, \beta \in \mathbb{C}$$

where the bar denotes complex conjugation. $\mathbf{U}$ transforms a complex projective point $\mathbf{z} = [z_1, z_2]^T$ and complex plane point $z$ by:

$$\mathbf{U} : \; \mathbf{z} \mapsto \mathbf{z}' = \mathbf{U}\mathbf{z} = \begin{bmatrix} \alpha & \beta \\ -\bar{\beta} & \bar{\alpha} \end{bmatrix} \begin{bmatrix} z_1 \\ z_2 \end{bmatrix} \tag{32}$$

$$\mathbf{\Phi}_{\mathbf{U}} : \; z \mapsto z' = \frac{\alpha z + \beta}{-\bar{\beta}z + \bar{\alpha}}, \;\; -\bar{\beta}z + \bar{\alpha} \neq 0 \tag{33}$$

These transformations are of importance as they act analogously to rotations of the unit sphere in $\mathbb{R}^3$. Specifically, for a 3x3 rotation matrix $\mathbf{R} \in SO(3)$ that rotates a unit vector $\mathbf{v} \in S^2$ as $\mathbf{v}' = \mathbf{R}\mathbf{v}$, there exists some $\mathbf{U}$ such that:

$$\mathbf{v}' = (\psi^{-1} \circ \mathbf{U} \circ \psi)(\mathbf{v}) \tag{34}$$

The exact relationship between $SU(2)$ and $SO(3)$ is made clearer by their relationships with unit quaternions $\mathbf{q} \in \mathbb{H}$ which also act as rotations in $\mathbb{R}^3$. The isomorphism between $SU(2)$ and unit quaternions is given as:

$$\mathbf{q} = w_q + x_q i + y_q j + z_q k, \quad w_q^2 + x_q^2 + y_q^2 + z_q^2 = 1, \quad w_q, x_q, y_q, z_q \in \mathbb{R}$$
$$\alpha = w_q + x_q i, \ \beta = y_q + z_q i \tag{35}$$

and the mapping of unit quaternions to special orthogonal matrices is given by:

$$\mathbf{R_q} = \begin{bmatrix} 1 - 2y_q^2 - 2z_q^2 & 2x_q y_q - 2w_q z_q & 2x_q z_q + 2w_q y_q \\ 2x_q y_q + 2w_q z_q & 1 - 2x_q^2 - 2z_q^2 & 2y_q z_q - 2w_q x_q \\ 2x_q z_q - 2w_q y_q & 2y_q z_q + 2w_q x_q & 1 - 2x_q^2 - 2y_q^2 \end{bmatrix} \tag{36}$$

Eq. (36) is the well-known 2-to-1 surjective mapping between quaternions and rotation matrices. By their isomorphism in Eq. (35), $SU(2)$ also has a similar surjective mapping with $SO(3)$, linking the three rotation representations. Note that the mapping given by Eq. (35) is not unique. Furthermore, special unitary matrices have the ability to act as rotations in $\mathbb{R}^3$ directly by first mapping points to 2x2 complex matrices. For a point $\mathbf{x} = (x, y, z) \in \mathbb{R}^3$:

$$\chi: \ \mathbf{x} \mapsto \mathbf{X} = \begin{bmatrix} xi & y + zi \\ -y + zi & -xi \end{bmatrix} \tag{37}$$

$$\chi(\mathbf{x}_1) \mapsto \mathbf{X}_1, \quad \chi(\mathbf{x}_2) \mapsto \mathbf{X}_2, \quad \mathbf{x}_1, \mathbf{x}_2 \in \mathbb{R}^3$$
$$\mathbf{X}_2 = \mathbf{U}\mathbf{X}_1\mathbf{U}^H, \quad \mathbf{U} \in SU(2) \tag{38}$$

Note if $||\mathbf{x}|| = 1$, $\chi(\mathbf{x}) \in SU(2)$. Also note that the map $\chi$ is not uniquely defined either.

Relatedly, Möbius transformations are general 2x2 complex projective matrices, characterized similarly by:

$$\mathbf{M} = \begin{bmatrix} \sigma & \xi \\ \gamma & \delta \end{bmatrix} \tag{39}$$

$$det(\mathbf{M}) \neq 0, \quad \sigma, \xi, \gamma, \delta \in \mathbb{C}$$

$$\mathbf{M}: \ \mathbf{z} \mapsto \mathbf{z}' = \mathbf{M}\mathbf{z} = \begin{bmatrix} \sigma & \xi \\ \gamma & \delta \end{bmatrix} \begin{bmatrix} z_1 \\ z_2 \end{bmatrix} \tag{40}$$

$$\mathbf{\Phi_M}: z \mapsto z' = \frac{\sigma z + \xi}{\gamma z + \delta}, \quad \gamma z + \delta \neq 0 \tag{41}$$

$$\mathbf{M} \sim \lambda \mathbf{M}, \quad \lambda \in \mathbb{C}, \ \lambda \neq 0 \tag{42}$$

Möbius transformations conformally map the complex projective plane onto itself. They are uniquely determined (up to scale) by their action on three independent points, and $SU(2)$ elements constitute a subset of them.

## B  Proofs and Derivations

### B.1  Proper Metric in Complex Projective Space

#### B.1.1  Derivation of Metric

Complex projective rays are equivalent if they are linearly dependent. We can test this condition by setting up the following constraint on complex vectors $\mathbf{z} = [z_1, z_2]^T$ and $\mathbf{p} = [p_1, p_2]^T$ for $z_1, z_2, p_1, p_2 \in \mathbb{C}$:

$$det\left(\begin{bmatrix} z_1 & p_1 \\ z_2 & p_2 \end{bmatrix}\right) = z_1 p_2 - z_2 p_1 = 0$$

For vectors $\mathbf{a} = (x_s, y_s, z_s), \mathbf{b} = (m_s, n_s, p_s) \in S^2$ (assume $\mathbf{a} \neq \mathbf{p}^*, \mathbf{b} \neq \mathbf{p}^*$) whose projections via $\psi$ (Eq. (30)) correspond to $\mathbf{z}$ and $\mathbf{p}$ respectively, we can show that testing the linear independence

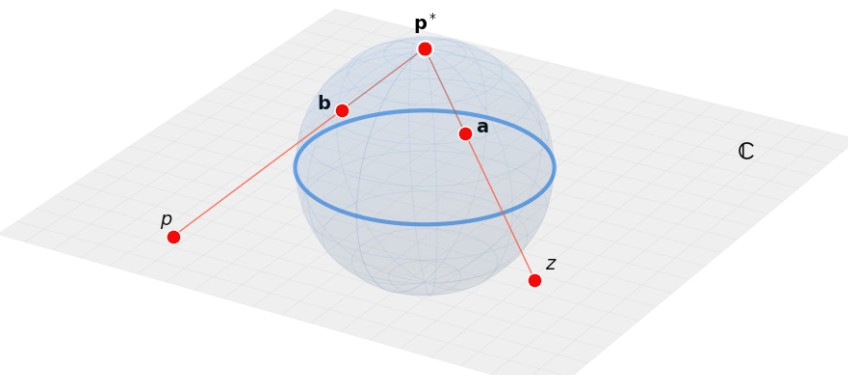

Figure 3: Visualization of a stereographic projection from the sphere ($S^2$) to the complex plane. The projection is performed by taking the line between $\mathbf{p}^*$ and each point and intersecting that line with the plane through the equator. The point $\mathbf{p}^*$ itself is mathematically mapped to infinity.

of complex vectors is in fact related to the chordal distance on a sphere:

$$\mathbf{z} = \lambda_1 \begin{bmatrix} x_s + y_s i \\ 1 + z_s \end{bmatrix}, \quad \mathbf{p} = \lambda_2 \begin{bmatrix} m_s + n_s i \\ 1 + p_s \end{bmatrix}, \quad \lambda_1, \lambda_2 \in \mathbb{C}, \ \lambda_1 \neq 0, \lambda_2 \neq 0$$

$$\left| det\left( \begin{bmatrix} z_1 & z_2 \\ p_1 & p_2 \end{bmatrix} \right) \right|^2 = |\lambda_1|^2 |\lambda_2|^2 |(1 + p_s)(x_s + y_s i) - (1 + z_s)(m_s + n_s i)|^2$$

$$= |\lambda_1|^2 |\lambda_2|^2 ((1 + p_s)^2 (x_s^2 + y_s^2) + (1 + z_s)^2 (m_s^2 + n_s^2) - 2(1 + p_s)(1 + z_s)(x_s m_s + y_s n_s))$$

$$= |\lambda_1|^2 |\lambda_2|^2 (1 + p_s)(1 + z_s)((1 + p_s)(1 - z_s) + (1 + z_s)(1 - p_s) - 2(x_s m_s + y_s n_s))$$

$$= |\lambda_1|^2 |\lambda_2|^2 (1 + p_s)(1 + z_s)(2 - 2(x_s m_s + y_s n_s + z_s p_s))$$

$$= |\lambda_1|^2 |\lambda_2|^2 (1 + p_s)(1 + z_s) ||\mathbf{a} - \mathbf{b}||^2$$

Notice that $|\lambda_1|^2 (1 + z_s) = \frac{|z_1|^2 + |z_2|^2}{2}$ and $|\lambda_2|^2 (1 + p_s) = \frac{|p_1|^2 + |p_2|^2}{2}$. Substituting this into our expression and rearranging, we arrive at the final expression for the equivalent distance metric in complex projective space as:

$$||\mathbf{a} - \mathbf{b}||^2 = \frac{4|z_1 p_2 - z_2 p_1|^2}{(|z_1|^2 + |z_2|^2)(|p_1|^2 + |p_2|^2)}$$

The last substitution may seem unnecessary at first; however, this form is more useful as it generalizes the metric to hold even when $\mathbf{a} = \mathbf{p}^*$ or $\mathbf{b} = \mathbf{p}^*$ (proof below). It also gives an intuitive interpretation that the spherical chordal distance is related to a type of "cross product" magnitude between the two projective rays' unit directions.

### B.1.2 PROOF OF METRIC FOR POINTS AT INFINITY

**Proposition 1** *If $\mathbf{a} = \mathbf{p}^*$ or $\mathbf{b} = \mathbf{p}^*$ in Eq. (4), the proper metric is still valid.*

*Proof* The squared distance between unit length points $\mathbf{a} = (x_s, y_s, z_s)$ and $\mathbf{b} = \mathbf{p}^* = (0, 0, -1)$ is:

$$||\mathbf{a} - \mathbf{b}||^2 = 2 - 2\mathbf{a}^T \mathbf{b} = 2(1 + z_s)$$

Using vectors $\mathbf{z} = \psi(\mathbf{a}) = \lambda_1 [x_s + y_s i, 1 + z_s]^T$, $\mathbf{p} = \psi(\mathbf{p}^*) = [\lambda_2, 0]^T$ with nonzero $\lambda_1, \lambda_2 \in \mathbb{C}$ and $\mathbf{a} \neq \mathbf{p}^*$, we can calculate the same quantity via the formula in Eq. (4):

$$\frac{4|z_1 p_2 - p_1 z_2|^2}{||\mathbf{z}||^2 ||\mathbf{p}||^2} = \frac{4|-\lambda_1 \lambda_2 (1 + z_s)|^2}{2|\lambda_1|^2 |\lambda_2|^2 (1 + z_s)} = 2(1 + z_s)$$

thus showing that the two formulas yield the same quantity. It is easy to see that Eq. (4) is symmetric, so the same result would hold if $\mathbf{a} = \mathbf{p}^*$ and $\mathbf{b} \neq \mathbf{p}^*$. If $\mathbf{a} = \mathbf{b} = \mathbf{p}^*$, we can see that $||\mathbf{a} - \mathbf{b}||^2$ is clearly 0. At the same time, the numerator of Eq. (4) would be 0 while the denominator is nonzero as the projective scalars $\lambda_i \neq 0$ for any valid complex projective point. Thus, both quantities are equal in that case as well, so the formula gives the spherical chordal distance between any two points on the sphere via their stereographic projections.

## B.2 Representation Derivations

### B.2.1 Derivation of 2-vec

For 3D vectors $\mathbf{b}_x, \mathbf{b}_y$ extracted from a model output representing predicted target $x$ and $y$ axes respectively, we apply the method from Section 3.3 in the unweighted case to arrive at an optimal rotation matrix (in the sense of Wahba's problem). We assume $\mathbf{b}_x \times \mathbf{b}_y \neq 0$. First, $\mathbf{b}_x$ and $\mathbf{b}_y$ must have the same norm for the method to be unweighted, so we transform $\mathbf{b}_y$ via $\mathbf{b}'_y = \sqrt{\frac{||\mathbf{b}_x||^2}{||\mathbf{b}_y||^2}} \mathbf{b}_y$. Since the reference points are constant ($\mathbf{a}_1 = (1, 0, 0), \mathbf{a}_2 = (0, 1, 0)$), we know that their normalized sum and difference vectors are $\mathbf{a}^+ = \frac{1}{\sqrt{2}}(1, 1, 0), \mathbf{a}^- = \frac{1}{\sqrt{2}}(1, -1, 0)$. Similarly, we create normalized sum and difference vectors for the target points as $\mathbf{b}^+ = \frac{\mathbf{b}_x + \mathbf{b}'_y}{||\mathbf{b}_x + \mathbf{b}'_y||}$ and $\mathbf{b}^- = \frac{\mathbf{b}_x - \mathbf{b}'_y}{||\mathbf{b}_x - \mathbf{b}'_y||}$. The optimal rotation aligns $\mathbf{a}^+$ to $\mathbf{b}^+$ and $\mathbf{a}^-$ to $\mathbf{b}^-$ noiselessly. This can be achieved because all the vectors have the same magnitude (normalizing to unit norm was found to be more stable than matching magnitudes like $\mathbf{b}'_y$) and because the sum and difference vectors are always orthogonal. Since rotation matrices naturally encode how an orthogonal coordinate frame transforms in their columns, we can construct the aligning rotation by joining the two rotations $\mathbf{R_a}$ and $\mathbf{R_b}$ which rotate the coordinate frame to the reference sum/difference vectors and target sum/difference vectors respectively:

$$\mathbf{R_a} = \begin{bmatrix} \mathbf{a}^+, & \mathbf{a}^-, & \mathbf{a}^+ \times \mathbf{a}^- \end{bmatrix}, \quad \mathbf{R_b} = \begin{bmatrix} \mathbf{b}^+, & \mathbf{b}^-, & \mathbf{b}^+ \times \mathbf{b}^- \end{bmatrix}$$

$$\mathbf{R} = \mathbf{R_b}\mathbf{R_a}^T = \begin{bmatrix} \frac{1}{\sqrt{2}}(\mathbf{b}^+ + \mathbf{b}^-), & \frac{1}{\sqrt{2}}(\mathbf{b}^+ - \mathbf{b}^-), & \mathbf{b}^- \times \mathbf{b}^+ \end{bmatrix}$$

Because the sum/difference vectors are orthogonal and have unit norm, $\mathbf{R_a}, \mathbf{R_b}, \mathbf{R} \in SO(3)$. Given the natural representation of coordinate transformations in rotation matrices, using the rotation matrix formulation was more appealing for the map than the quaternion formulation in Eq. (22). It also provided a more direct comparison with the Gram-Schmidt map. Nonetheless, the core insight was derived from the original linear constraints on quaternion parameters. The unweighted method was chosen for its geometric and computational simplicity, but a weighted version of the map incorporating the magnitudes of $\mathbf{b}_x, \mathbf{b}_y$ can be similarly formulated from Eq. (21).

### B.2.2 Derivation of QuadMobius Formulas

Following the algorithm in Section 2.2, we normalize a 2x2 complex projective matrix $\mathbf{M}$ by its determinant and find the nearest unitary matrix, which by Appendix B.3.1 is special unitary. The following are two different approaches to impelement this. We assume $\mathbf{M}$ has full rank.

**Linear Algebra** Instead of normalizing $\mathbf{M}$ directly, we take a more streamlined approach by utilizing the properties of polar decomposition and determinant. We express $det(\mathbf{M})$ in polar form as $re^{i\theta}$ with $r = |det(\mathbf{M})| \in \mathbb{R}, r > 0$ and $e^{i\theta} = \frac{det(\mathbf{M})}{|det(\mathbf{M})|}$ lying on the unit circle. For polar decomposition $\mathbf{M} = \mathbf{Q}\mathbf{P}$ with unitary matrix $\mathbf{Q}$ and positive definite Hermitian matrix $\mathbf{P}$, we have $det(\mathbf{M}) = det(\mathbf{Q})det(\mathbf{P})$. Because $\mathbf{Q}$ is unitary, $|det(\mathbf{Q})| = 1$, and because $\mathbf{P}$ is positive definite Hermitian, $det(\mathbf{P})$ is real and nonnegative. It follows then that $det(\mathbf{Q}) = e^{i\theta}$ and $det(\mathbf{P}) = r$. To normalize $\mathbf{M}$, we typically multiply it by a nonzero scalar $\lambda \in \mathbb{C}$. For polar decomposition to remain valid under this scaling, $\lambda$ must distribute as $\lambda\mathbf{M} = \left(\frac{\lambda}{|\lambda|}\mathbf{Q}\right)(|\lambda|\mathbf{P})$, meaning that only the phase of $\lambda$ affects the unitary factor. Since the unitary factor $\mathbf{Q}$ is the nearest unitary matrix to $\mathbf{M}$ in the Frobenius sense, the final solution is just $\frac{\lambda}{|\lambda|}\mathbf{Q}$ such that $det(\frac{\lambda}{|\lambda|}\mathbf{Q}) = 1$ to be special unitary. We can therefore reverse the order and first compute $\mathbf{Q}$ before normalizing its determinant. We find a scalar $\lambda'$ such that $det(\lambda'\mathbf{Q}) = \lambda'^2 det(\mathbf{Q}) = 1$ (since $\mathbf{Q}$ is 2x2) for $|\lambda'| = 1$. We can easily solve $\lambda' = det(\mathbf{Q})^{-\frac{1}{2}}$. Since $\mathbf{Q} = \mathbf{U}\mathbf{V}^H$ from SVD ($\mathbf{M} = \mathbf{U}\mathbf{\Sigma}\mathbf{V}^H$) and $|det(\mathbf{Q})| = 1$, we can rewrite our expression simply as $\sqrt{det(\mathbf{U}\mathbf{V}^H)}\mathbf{U}\mathbf{V}^H$. If $\mathbf{M}$ is singular, there is no unique solution as SVD is no longer unique. This formula may still be used in practice with a specific SVD.

**Algebraic** First, we can normalize $\mathbf{M}$ to $\mathbf{M}' = det(\mathbf{M})^{-\frac{1}{2}}\mathbf{M}$ such that $det(\mathbf{M}') = 1$. Next, we can utilize the isomorphism between $SU(2)$ and quaternions in Eq. (35) to algebraically solve for the nearest special unitary matrix. It's easy to verify that the unitary matrix $\mathbf{Q}$ that minimizes

the Frobenius distance to $\mathbf{M}'$ maximizes $\Re(Tr(\mathbf{M}'^H\mathbf{Q}))$ where $\Re(\cdot)$ denotes the real part. From Appendix B.3.1, we know that $\mathbf{Q}$ will be special unitary. Thus, we can express the optimization problem (using symbols from Eqs. (31) and (39)) as:

$$\max_{\mathbf{Q}\in SU(2)} \Re(Tr(\mathbf{M}^H\mathbf{Q})) = \Re(\overline{\sigma}\alpha + \overline{\xi}\beta + \overline{\delta}\alpha - \overline{\gamma}\beta)$$

$$= \max_{||\mathbf{q}||=1} (\Re(\sigma) + \Re(\delta))w_q + (\Im(\sigma) - \Im(\delta))x_q + (\Re(\xi) - \Re(\gamma))y_q + (\Im(\xi) + \Im(\gamma))z_q$$

for quaternion $\mathbf{q} = w_q + x_q i + y_q j + z_q k$ and $\Im(\cdot)$ denoting the imaginary part. For $\mathbf{q}$ to be a valid rotation, it must have unit norm. Thus, the optimization problem can be rephrased as finding the unit norm vector whose dot product with the coefficients of the quaternion parameters above is maximized. The solution is trivially obtained by the unit norm vector in the direction of those coefficients. Using Eq. (35) again, we can express the solution as:

$$\tilde{\mathbf{q}} = (\Re(\sigma) + \Re(\delta)) + (\Im(\sigma) - \Im(\delta))i + (\Re(\xi) - \Re(\gamma))j + (\Im(\xi) + \Im(\gamma))k$$

$$\tilde{\alpha} = \sigma + \overline{\delta}, \quad \tilde{\beta} = \xi - \overline{\gamma}$$

$$\mathbf{Q} \sim \mathbf{M}' + adj(\mathbf{M}')^H$$

where tilde denotes unnormalized parameters and $adj(\cdot)$ denotes the adjugate. We can normalize the parameters by dividing $\tilde{\alpha}$ and $\tilde{\beta}$ by $\sqrt{|\tilde{\alpha}|^2 + |\tilde{\beta}|^2} = \sqrt{|\sigma + \overline{\delta}|^2 + |\xi - \overline{\gamma}|^2} = \sqrt{Tr(\mathbf{M}'^H\mathbf{M}') + 2\Re(det(\mathbf{M}'))} = \sqrt{Tr(\mathbf{M}'^H\mathbf{M}') + 2}$. Since that factor is real and distributes linearly through $\tilde{\alpha}$ and $\tilde{\beta}$ to the elements of $\mathbf{M}'$, we can efficiently combine this normalization factor into the original normalization factor of $det(\mathbf{M})^{-\frac{1}{2}}$ in the first step. The combined normalization factor can be written as:

$$\frac{1}{\sqrt{det(\mathbf{M})}}\frac{1}{\sqrt{Tr(\mathbf{M}'^H\mathbf{M}') + 2}} = \frac{1}{\sqrt{det(\mathbf{M})}}\frac{1}{\sqrt{\frac{Tr(\mathbf{M}^H\mathbf{M})}{|det(\mathbf{M})|} + 2}}$$

$$= \sqrt{\frac{|det(\mathbf{M})|}{det(\mathbf{M})(Tr(\mathbf{M}^H\mathbf{M}) + 2|det(\mathbf{M})|)}} = \sqrt{\frac{\overline{det(\mathbf{M})}}{|det(\mathbf{M})|(Tr(\mathbf{M}^H\mathbf{M}) + 2|det(\mathbf{M})|)}}$$

Applying this normalization factor to $\mathbf{M}$ to obtain $\mathbf{M}^*$ will ensure that $\mathbf{M}^* + adj(\mathbf{M}^*)^H \in SU(2)$.

## B.3 NEAREST UNITARY MATRIX

### B.3.1 PROOF OF NEAREST SPECIAL UNITARY MATRIX

**Proposition 2** *If Möbius transformation* $\mathbf{M}$ *has* $det(\mathbf{M}) = 1$, *the nearest unitary matrix to* $\mathbf{M}$ *in the Frobenius sense is special unitary.*

*Proof* $\mathbf{M}$ has a singular value decomposition given as $\mathbf{M} = \mathbf{U}\mathbf{\Sigma}\mathbf{V}^H$ where $\mathbf{U}$ and $\mathbf{V}$ are unitary matrices and $\mathbf{\Sigma}$ is a diagonal matrix with singular values. The determinant of $\mathbf{M}$ can be expressed as:

$$det(\mathbf{M}) = det(\mathbf{U})det(\mathbf{\Sigma})det(\mathbf{V}^H) \tag{43}$$

by product rule of determinants. Multiplying both sides by their complex conjugates, we obtain:

$$|det(\mathbf{M})|^2 = |det(\mathbf{U})|^2|det(\mathbf{\Sigma})|^2|det(\mathbf{V}^H)|^2$$

Since $\mathbf{U}$ and $\mathbf{V}^H$ are unitary matrices, the magnitude of their determinant is 1, so the expression simplifies to:

$$|det(\mathbf{M})|^2 = |det(\mathbf{\Sigma})|^2 \implies |det(\mathbf{M})| = |det(\mathbf{\Sigma})|$$

because the determinant magnitudes are real and nonnegative. Since $\mathbf{\Sigma}$ is a diagonal matrix with real, nonnegative elements, its determinant is simply the product of its diagonal entries and is in turn real and nonnegative. If $det(\mathbf{M}) = 1$, then $|det(\mathbf{\Sigma})| = det(\mathbf{\Sigma}) = 1$. Coming back to the first expression, we can now write:

$$det(\mathbf{M}) = det(\mathbf{U})det(\mathbf{V}^H) = det(\mathbf{U}\mathbf{V}^H) = 1$$

It is known that closest unitary matrix to $\mathbf{M}$ in the Frobenius sense is the unitary part of polar decomposition (Keller, 1975) which can be computed by $\mathbf{U}\mathbf{V}^H$. From above, we can see that $det(\mathbf{U}\mathbf{V}^H) = 1$ which means that $\mathbf{U}\mathbf{V}^H$ is special unitary by definition.

In noiseless situations, $\mathbf{\Sigma}$ is observed to be the identity matrix if $det(\mathbf{M}) = 1$. As noise is added, the diagonal elements of $\mathbf{\Sigma}$ drift from 1, so $\mathbf{\Sigma}$ encodes a notion of how close a Möbius transformation's action is to a rotation or how much noise the problem contains, making it a candidate for optimization.

### B.3.2   DERIVATION OF NEAREST UNITARY MATRIX DERIVATIVE

The nearest unitary matrix in the Frobenius sense to a complex square matrix $\mathbf{M}$ is given by the unitary factor $\mathbf{Q}$ of its polar decomposition $\mathbf{M} = \mathbf{QP}$ where $\mathbf{P}$ is a positive semidefinite Hermitian matrix (Keller, 1975). We can find the derivative of $\mathbf{Q}$ with respect to the elements of $\mathbf{M}$ by taking the derivative of both sides of the polar decomposition:

$$dM = d(\mathbf{QP})$$
$$dM = (d\mathbf{Q})\mathbf{P} + \mathbf{Q}(d\mathbf{P})$$
$$\mathbf{Q}^H(d\mathbf{M}) = \mathbf{Q}^H(d\mathbf{Q})\mathbf{P} + d\mathbf{P}$$

Taking the conjugate transpose of both sides and subtracting the two statements:

$$(d\mathbf{M}^H)\mathbf{Q} = \mathbf{P}^H(d\mathbf{Q}^H)\mathbf{Q} + d\mathbf{P}^H$$
$$\mathbf{Q}^H(d\mathbf{M}) - (d\mathbf{M}^H)\mathbf{Q} = \mathbf{Q}^H(d\mathbf{Q})\mathbf{P} - \mathbf{P}^H(d\mathbf{Q}^H)\mathbf{Q} + (d\mathbf{P} - d\mathbf{P}^H)$$

We observe that because $\mathbf{P}$ is Hermitian for all values of $\mathbf{M}$, $d\mathbf{P}$ must also be Hermitian, so the last term cancels out. Furthermore, we can deduce the following from definition of unitary matrices:

$$\mathbf{Q}^H\mathbf{Q} = \mathbf{I}$$
$$(d\mathbf{Q}^H)\mathbf{Q} + \mathbf{Q}^H(d\mathbf{Q}) = 0$$
$$(d\mathbf{Q}^H)\mathbf{Q} = -\mathbf{Q}^H(d\mathbf{Q})$$

implying that $(d\mathbf{Q}^H)\mathbf{Q}$ is skew-Hermitian. Denoting $\mathbf{X} = \mathbf{Q}^H(d\mathbf{Q})$ and $\mathbf{C} = \mathbf{Q}^H(d\mathbf{M}) - (d\mathbf{M}^H)\mathbf{Q}$, we can now write:

$$\mathbf{C} = \mathbf{XP} + \mathbf{PX}$$

which takes the form of a Sylvester equation. Since $\mathbf{P}$ is Hermitian, it admits a diagonalization $\mathbf{P} = \mathbf{Y}\mathbf{\Lambda}\mathbf{Y}^H$, where $\mathbf{Y}$ is unitary and $\mathbf{\Lambda}$ is a diagonal matrix of eigenvalues of $\mathbf{P}$:

$$\mathbf{C} = \mathbf{XY}\mathbf{\Lambda}\mathbf{Y}^H + \mathbf{Y}\mathbf{\Lambda}\mathbf{Y}^H\mathbf{X}$$
$$\mathbf{Y}^H\mathbf{C}\mathbf{Y} = (\mathbf{Y}^H\mathbf{XY})\mathbf{\Lambda} + \mathbf{\Lambda}(\mathbf{Y}^H\mathbf{XY})$$

The right hand side has the same term $\mathbf{Y}^H\mathbf{XY}$ multiplied on the left and right respectively by diagonal matrix $\mathbf{\Lambda}$. As such, we can equivalently express the result as follows in order to solve for $\mathbf{X}$ and ultimately $d\mathbf{Q}$:

$$\mathbf{Y}^H\mathbf{C}\mathbf{Y} = (diag(\mathbf{\Lambda}) \oplus diag(\mathbf{\Lambda})) \odot (\mathbf{Y}^H\mathbf{XY})$$
$$\mathbf{Y}^H\mathbf{XY} = \frac{\mathbf{Y}^H\mathbf{C}\mathbf{Y}}{diag(\mathbf{\Lambda}) \oplus diag(\mathbf{\Lambda})}$$
$$\mathbf{X} = \mathbf{Y}\Big(\frac{\mathbf{Y}^H\mathbf{C}\mathbf{Y}}{diag(\mathbf{\Lambda}) \oplus diag(\mathbf{\Lambda})}\Big)\mathbf{Y}^H$$
$$d\mathbf{Q} = \mathbf{QY}\Big(\frac{\mathbf{Y}^H(\mathbf{Q}^H(d\mathbf{M}) - (d\mathbf{M}^H)\mathbf{Q})\mathbf{Y}}{diag(\mathbf{\Lambda}) \oplus diag(\mathbf{\Lambda})}\Big)\mathbf{Y}^H$$

where $\oplus$ denotes an outer sum operation, $\odot$ denotes Hadamard multiplication (element-wise), the division is Hadamard division (element-wise), and $diag(\cdot)$ is a vector formed from the diagonal elements of the matrix . Note that this solution is only properly defined if $\mathbf{M}$ is nonsingular (i.e. $\mathbf{\Lambda}$ has full rank). Otherwise, the polar decomposition is not unique and neither is its derivative. In practice, we choose to replace any instances of division by 0 in the result above with multiplications by 0 as a specific solution.

### B.4 TWO-POINT SOLUTIONS

#### B.4.1 PROOF OF WEIGHTED CASE

**Proposition 3** *Let $\mathbf{a}_i$ and $\mathbf{b}_i$ represent the reference and target points respectively and $\mathbf{k}_a = \mathbf{a}_1 \times \mathbf{a}_2$ and $\mathbf{k}_b = \mathbf{b}_1 \times \mathbf{b}_2$. For $n = 2$ points, $\mathbf{k}_a \neq \mathbf{0}$, and $\mathbf{k}_b \neq \mathbf{0}$, the optimal rotation to Wahba's problem is given as the weighted average (in the Frobenius sense) between two rotations $\mathbf{R}_1$ and $\mathbf{R}_2$ defined by $\mathbf{R}_i \mathbf{a}_i = \mathbf{b}_i$ and $\mathbf{R}_i \frac{\mathbf{k}_a}{||\mathbf{k}_a||} = \frac{\mathbf{k}_b}{||\mathbf{k}_b||}$.*

*Lemma: If all points lie in the plane z=0 and $\mathbf{k}_a \neq 0, \mathbf{k}_b \neq 0$, and $\mathbf{k}_a \cdot \mathbf{k}_b > 0$, the optimal rotation is a rotation around the z-axis.*

Since all points lie in the plane $z = 0$, the last column and row of $\mathbf{B}$ (Eq. (2)) are zero. As a result, the last column and row of $\mathbf{B}\mathbf{B}^T$ and $\mathbf{B}^T\mathbf{B}$ are also zero, so they both have a kernel vector of $(0, 0, 1)$. For the SVD of $\mathbf{B}$ given as $\mathbf{U}\mathbf{\Sigma}\mathbf{V}^T$, the optimal rotation $\mathbf{R}$ (via Markley (1987)) can take the form:

$$\mathbf{R} = \begin{bmatrix} \cdot & \cdot & 0 \\ \cdot & \cdot & 0 \\ 0 & 0 & 1 \end{bmatrix} \begin{bmatrix} 1 & 0 & 0 \\ 0 & 1 & 0 \\ 0 & 0 & det(\mathbf{U})det(\mathbf{V}) \end{bmatrix} \begin{bmatrix} \cdot & \cdot & 0 \\ \cdot & \cdot & 0 \\ 0 & 0 & 1 \end{bmatrix}$$

where $det(\mathbf{U})det(\mathbf{V})$ is either 1 or -1 since $\mathbf{U}$ and $\mathbf{V}$ are orthogonal matrices. Thus, the last column and row of $\mathbf{R}$ are both $(0, 0, 1)$ or $(0, 0, -1)$. In order for $\mathbf{R}$ to be a valid rotation matrix, the remaining upper 2x2 submatrix must be an orthogonal matrix which can be generated by a single parameter $\theta$. Furthermore, the sign of the bottom right corner element of $\mathbf{R}$ must be the same as the determinant of the upper 2x2 submatrix for $det(\mathbf{R}) = 1$. These conditions reduce $\mathbf{R}$ to one of the two general forms:

$$\begin{bmatrix} cos(\theta_1) & -sin(\theta_1) & 0 \\ sin(\theta_1) & cos(\theta_1) & 0 \\ 0 & 0 & 1 \end{bmatrix}, \quad \begin{bmatrix} cos(\theta_2) & sin(\theta_2) & 0 \\ sin(\theta_2) & -cos(\theta_2) & 0 \\ 0 & 0 & -1 \end{bmatrix}$$

We denote the former as $\mathbf{R}_{SO}$ and the latter as $\mathbf{R}_O$. The optimal solution to Wahba's problem maximizes the gain function $Tr(\mathbf{R}\mathbf{B}^T)$ Lourakis and Terzakis (2018). This quantity for both forms can be expressed as below:

$$Tr(\mathbf{R}_{SO}\mathbf{B}^T) = \lambda_{1,1}cos(\theta_1) + \lambda_{1,2}sin(\theta_1)$$
$$Tr(\mathbf{R}_O\mathbf{B}^T) = \lambda_{2,1}cos(\theta_2) + \lambda_{2,2}sin(\theta_2)$$
$$\lambda_{1,1} = \mathbf{B}_{1,1} + \mathbf{B}_{2,2}, \ \lambda_{1,2} = \mathbf{B}_{2,1} - \mathbf{B}_{1,2}$$
$$\lambda_{2,1} = \mathbf{B}_{1,1} - \mathbf{B}_{2,2}, \ \lambda_{2,2} = \mathbf{B}_{2,1} + \mathbf{B}_{1,2}$$

The gain function in both cases is the dot product between $(\lambda_{i,1}, \lambda_{i,2})$ and $(cos(\theta_i), sin(\theta_i))$. Its maximum value (subject to the constraint $cos(\theta_i)^2 + sin(\theta_i)^2 = 1$) is obtained by the unit vector aligned with $(\lambda_{i,1}, \lambda_{i,2})$, i.e.:

$$cos(\theta_i) = \frac{\lambda_{i,1}}{\sqrt{\lambda_{i,1}^2 + \lambda_{i,2}^2}}, \ \ sin(\theta_i) = \frac{\lambda_{i,2}}{\sqrt{\lambda_{i,1}^2 + \lambda_{i,2}^2}}$$

Substituting this back into the gain function, we see that the optimal value is simply the magnitude of $(\lambda_{i,1}, \lambda_{i,2})$:

$$Tr(\mathbf{R}_{SO}\mathbf{B}^T) = \sqrt{\lambda_{1,1}^2 + \lambda_{1,2}^2}, \ \ Tr(\mathbf{R}_O\mathbf{B}^T) = \sqrt{\lambda_{2,1}^2 + \lambda_{2,2}^2}$$

Since the square root function is monotonically increasing, the larger of the two radicands corresponds to the larger gain value. We can compare them directly by taking their difference:

$$(\lambda_{1,1}^2 + \lambda_{1,2}^2) - (\lambda_{2,1}^2 + \lambda_{2,2}^2) = 4w_1 w_2 (\mathbf{k}_a \cdot \mathbf{k}_b)$$

where $w_i$ are the weights. Since the weights are positive and the cross products are assumed nonzero, the quantity above is positive when $\mathbf{k}_a$ and $\mathbf{k}_b$ point in the same direction and negative otherwise. Thus, when the cross products of the reference and target sets are aligned, $\mathbf{R}_{SO}$ corresponds to the larger gain value and is the optimal rotation. It takes the form of a rotation about the z-axis.

*Proof* We assume that all points lie in the plane $z = 0$ and that the cross product of the reference and target sets are nonzero and are aligned. This will be generalized later. We construct rotations $\mathbf{R}_1$ and $\mathbf{R}_2$ to be rotations about the z-axis that align $\mathbf{a}_1$ to $\mathbf{b}_1$ and $\mathbf{a}_2$ to $\mathbf{b}_2$ respectively. Since the input points have unit length and the vector norm is rotationally invariant, we can rewrite the loss function as:

$$w_1||\mathbf{b}_1 - \mathbf{R}\mathbf{a}_1||^2 + w_2||\mathbf{b}_2 - \mathbf{R}\mathbf{a}_2||^2$$
$$= w_1||\mathbf{a}_1 - \mathbf{R}_1^T\mathbf{R}\mathbf{a}_1||^2 + w_2||\mathbf{a}_2 - \mathbf{R}_2^T\mathbf{R}\mathbf{a}_2||^2$$
$$= w_1||(\mathbf{I} - \mathbf{R}_1^T\mathbf{R})\mathbf{a}_1||^2 + w_2||(\mathbf{I} - \mathbf{R}_2^T\mathbf{R})\mathbf{a}_2||^2$$
$$= w_1\mathbf{a}_1^T(\mathbf{I} - \mathbf{R}_1^T\mathbf{R})^T(\mathbf{I} - \mathbf{R}_1^T\mathbf{R})\mathbf{a}_1 + w_2\mathbf{a}_2^T(\mathbf{I} - \mathbf{R}_2^T\mathbf{R})^T(\mathbf{I} - \mathbf{R}_2^T\mathbf{R})\mathbf{a}_2$$
$$= 2(w_1 + w_2) - 2w_1\mathbf{a}_1^T\mathbf{R}_1^T\mathbf{R}\mathbf{a}_1 - 2w_2\mathbf{a}_2^T\mathbf{R}_2^T\mathbf{R}\mathbf{a}_2$$

using the fact $\mathbf{a}_i^T\mathbf{R}_i^T\mathbf{R}\mathbf{a}_i = \mathbf{a}_i^T\mathbf{R}^T\mathbf{R}_i\mathbf{a}_i$. Under our assumptions, the lemma establishes that the optimal rotation $\mathbf{R}$ is a rotation about the z-axis. Since both $\mathbf{R}_1$ and $\mathbf{R}_2$ are also rotations about the z-axis, we can easily verify that the products $\mathbf{R}_1^T\mathbf{R}$ and $\mathbf{R}_2^T\mathbf{R}$ are rotations about the z-axis as well. Using Rodrigues' rotation formula, we can expand the term below as follows:

$$\mathbf{a}_1^T\mathbf{R}_1^T\mathbf{R}\mathbf{a}_1 = \mathbf{a}_1 \cdot (cos(\phi)\mathbf{a}_1 + sin(\phi)\mathbf{k} \times \mathbf{a}_1 + (1 - cos(\phi))(\mathbf{k} \cdot \mathbf{a}_1)\mathbf{k})$$
$$= cos(\phi) + sin(\phi)(\mathbf{a}_1 \cdot (\mathbf{k} \times \mathbf{a}_1)) = cos(\phi)$$

where $\phi$ is the angle of rotation of $\mathbf{R}_1^T\mathbf{R}$ and $\mathbf{k} = [0, 0, 1]^T$ is the axis of rotation. The simple result is due to the fact that $\mathbf{a}_1$ is orthogonal to the axis of rotation and has unit length. On the other hand, we note that the Frobenius norm between $\mathbf{R}_1$ and $\mathbf{R}$ computes the following:

$$||\mathbf{R}_1 - \mathbf{R}||_F^2 = Tr((\mathbf{R}_1 - \mathbf{R})^T(\mathbf{R}_1 - \mathbf{R}))$$
$$= 6 - 2Tr(\mathbf{R}_1^T\mathbf{R})$$
$$= 6 - 2Tr(cos(\phi)\mathbf{I} + sin(\phi)[\mathbf{k}]_\times + (1 - cos(\phi))\mathbf{k}\mathbf{k}^T)$$
$$= 6 - 6cos(\phi) - 2(1 - cos(\phi)) = 4 - 4cos(\phi)$$
$$cos(\phi) = 1 - \frac{1}{4}||\mathbf{R}_1 - \mathbf{R}||_F^2$$

The expansion of $\mathbf{R}_1^T\mathbf{R}_1$ above is due to the axis-angle formula for rotation matrices where $[\mathbf{k}]_\times$ denotes the traceless skew-symmetric matrix formed from $\mathbf{k}$ representing a vector cross product. Deriving a similar result for $\mathbf{a}_2^T\mathbf{R}_2^T\mathbf{R}\mathbf{a}_2$ and plugging both back into our reformulated loss function, we can rewrite it as:

$$2(w_1 + w_2) - 2w_1(1 - \frac{1}{4}||\mathbf{R}_1 - \mathbf{R}||_F^2) - 2w_2(1 - \frac{1}{4}||\mathbf{R}_2 - \mathbf{R}||_F^2)$$
$$= \frac{1}{2}w_1||\mathbf{R}_1 - \mathbf{R}||_F^2 + \frac{1}{2}w_2||\mathbf{R}_2 - \mathbf{R}||_F^2$$

Through this expression, we can see that the rotation $\mathbf{R}$ which minimized our original loss is exactly the rotation that represents the weighted average in the Frobenius sense between $\mathbf{R}_1$ and $\mathbf{R}_2$ as specified in Markley et al. (2007). The uniform factor of $\frac{1}{2}$ is irrelevant to the optimization.

Now we generalize the result. Starting from the assumed configuration, we can extend it to general configurations by applying arbitrary rotations $\mathbf{R}_a$ and $\mathbf{R}_b$ to the reference and target points respectively, transforming them into $\mathbf{a}_i'$ and $\mathbf{b}_i'$. In this new coordinate frame, the rotation matrix $\mathbf{R}'$ is related to the original optimal matrix $\mathbf{R}$ as shown below:

$$\sum_i w_i||\mathbf{b}_i - \mathbf{R}\mathbf{a}_i||^2 = \sum_i w_i||\mathbf{R}_b\mathbf{b}_i - \mathbf{R}_b\mathbf{R}\mathbf{a}_i||^2$$
$$= \sum_i w_i||\mathbf{R}_b\mathbf{b}_i - \mathbf{R}_b\mathbf{R}(\mathbf{R}_a^T\mathbf{R}_a)\mathbf{a}_i||^2 = \sum_i w_i||\mathbf{b}_i' - (\mathbf{R}_b\mathbf{R}\mathbf{R}_a^T)\mathbf{a}_i'||^2$$
$$\mathbf{R}' = \mathbf{R}_b\mathbf{R}\mathbf{R}_a^T$$

Because the vector norm is invariant under rotation, the optimal loss value remains unchanged across all coordinate frames. Since the optimal value from the original coordinate frame is preserved

above, $\mathbf{R}'$ represents the optimal rotation in the new frame. Furthermore, the Frobenius norm is also rotation-invariant, so we can apply the required rotations to estimate $\mathbf{R}'$ as follows:

$$\sum_i w_i ||\mathbf{R}_i - \mathbf{R}||_F^2 = \sum_i w_i ||\mathbf{R}_b \mathbf{R}_i \mathbf{R}_a^T - \mathbf{R}_b \mathbf{R} \mathbf{R}_a^T||_F^2$$

$$= \sum_i w_i ||\mathbf{R}_b \mathbf{R}_i \mathbf{R}_a^T - \mathbf{R}'||_F^2$$

$$\mathbf{R}'_1 = \mathbf{R}_b \mathbf{R}_1 \mathbf{R}_a^T, \ \ \mathbf{R}'_2 = \mathbf{R}_b \mathbf{R}_2 \mathbf{R}_a^T$$

Thus, in the general case, the optimal rotation is given by the weighted average rotation between $\mathbf{R}'_1$ and $\mathbf{R}'_2$. We can uniquely identify those rotations with at least two linearly independent points they transform. Starting with the reference and target sets:

$$\mathbf{R}_i \mathbf{a}_i \equiv \mathbf{b}_i$$

$$\mathbf{R}_b \mathbf{R}_i (\mathbf{R}_a^T \mathbf{R}_a) \mathbf{a}_i = \mathbf{R}_b \mathbf{b}_i$$

$$\mathbf{R}'_i \mathbf{a}'_i = \mathbf{b}'_i$$

Each rotation still aligns their respective reference point to their target point. Furthermore, in our original coordinate frame, $\mathbf{k}_a$ and $\mathbf{k}_b$ are aligned and are parallel or antiparallel to $\mathbf{R}_i$'s axis of rotation (z-axis), so they are unchanged by $\mathbf{R}_i$. As a result:

$$\mathbf{R}_i \frac{\mathbf{k}_a}{||\mathbf{k}_a||} = \frac{\mathbf{k}_b}{||\mathbf{k}_b||}$$

$$\mathbf{R}_b \mathbf{R}_i (\mathbf{R}_a^T \mathbf{R}_a) \frac{\mathbf{k}_a}{||\mathbf{k}_a||} = \mathbf{R}_b \frac{\mathbf{k}_b}{||\mathbf{k}_b||}$$

$$\mathbf{R}'_i \frac{\mathbf{R}_a (\mathbf{a}_1 \times \mathbf{a}_2)}{||\mathbf{R}_a (\mathbf{a}_1 \times \mathbf{a}_2)||} = \frac{\mathbf{R}_b (\mathbf{b}_1 \times \mathbf{b}_2)}{||\mathbf{R}_b (\mathbf{b}_1 \times \mathbf{b}_2)||}$$

$$\mathbf{R}'_i \frac{\mathbf{a}'_1 \times \mathbf{a}'_2}{||\mathbf{a}'_1 \times \mathbf{a}'_2||} = \frac{\mathbf{b}'_1 \times \mathbf{b}'_2}{||\mathbf{b}'_1 \times \mathbf{b}'_2||}$$

due to rotations distributing over the cross product. Thus, we can identify $\mathbf{R}'_1$ and $\mathbf{R}'_2$ as the rotations that align their corresponding reference point to their target point along with the cross products of the reference and target sets. As the cross products are assumed nonzero and are orthogonal to their respective point set, the two points aligned by each rotation are always independent and therefore uniquely define the rotations. As shown, the optimal rotation is the weighted average in the Frobenius sense between them.

### B.4.2 Proof of Unweighted Case

**Proposition 4** *Let $\mathbf{a}_i$, $\mathbf{b}_i$, and $w_i$ represent the reference points, target points, and weights respectively. Given $n = 2$ points, $w_1 = w_2$, $\mathbf{a}_1 \times \mathbf{a}_2 \neq \mathbf{0}$, and $\mathbf{b}_1 \times \mathbf{b}_2 \neq \mathbf{0}$, the optimal rotation to Wahba's problem is given by the unique rotation $\mathbf{R}$ defined by $\mathbf{R}(\frac{\mathbf{a}_1 + \mathbf{a}_2}{||\mathbf{a}_1 + \mathbf{a}_2||}) = \frac{\mathbf{b}_1 + \mathbf{b}_2}{||\mathbf{b}_1 + \mathbf{b}_2||}$ and $\mathbf{R}(\frac{\mathbf{a}_1 - \mathbf{a}_2}{||\mathbf{a}_1 - \mathbf{a}_2||}) = \frac{\mathbf{b}_1 - \mathbf{b}_2}{||\mathbf{b}_1 - \mathbf{b}_2||}$.*

*Proof* For two 3D unit vectors $\mathbf{v}_1$ and $\mathbf{v}_2$, we introduce the following notation and easily verifiable results:

$$\tilde{\mathbf{v}}^- \equiv \mathbf{v}_1 - \mathbf{v}_2, \ \ \tilde{\mathbf{v}}^+ \equiv \mathbf{v}_1 + \mathbf{v}_2$$

$$\mathbf{v}^- = \frac{\tilde{\mathbf{v}}^-}{||\tilde{\mathbf{v}}^-||}, \ \ \mathbf{v}^+ = \frac{\tilde{\mathbf{v}}^+}{||\tilde{\mathbf{v}}^+||}$$

$$\tilde{\mathbf{v}}^- \cdot \tilde{\mathbf{v}}^+ = 0$$

$$\mathbf{v}_1 \cdot \tilde{\mathbf{v}}^+ = \mathbf{v}_2 \cdot \tilde{\mathbf{v}}^+$$

$$\tilde{\mathbf{v}}^- \times \tilde{\mathbf{v}}^+ = 2(\mathbf{v}_1 \times \mathbf{v}_2)$$

$$\mathbf{v}_1 \times \mathbf{v}_2 \neq \mathbf{0} \implies \tilde{\mathbf{v}}^- \neq \mathbf{0}, \ \tilde{\mathbf{v}}^+ \neq \mathbf{0}$$

If $\mathbf{v}_1 \times \mathbf{v}_2 \neq \mathbf{0}$, then the two vectors $\mathbf{v}^-$ and $\mathbf{v}^+$ are well-defined and form an orthonormal basis for the plane spanned by $\mathbf{v}_1$ and $\mathbf{v}_2$. Consequently, $\mathbf{v}^-$ and $\mathbf{v}^+$ created from one pair of linearly independent unit vectors can be perfectly aligned with those created from another pair.

With $\mathbf{a}_1 \times \mathbf{a}_2 \neq \mathbf{0}, \mathbf{b}_1 \times \mathbf{b}_2 \neq \mathbf{0}$, we initially assume that the points are configured such that they all lie in the plane $z = 0$ and that $\mathbf{a}^+ = \mathbf{b}^+$ and $\mathbf{a}^- = \mathbf{b}^-$. This is generalized later. For this configuration, we note the following:

$$\mathbf{a}_1 \times \mathbf{a}_2 = \frac{1}{2}(\tilde{\mathbf{a}}^- \times \tilde{\mathbf{a}}^+)$$

$$= \frac{1}{2}||\tilde{\mathbf{a}}^-||||\tilde{\mathbf{a}}^+||(\mathbf{a}^- \times \mathbf{a}^+) = \frac{1}{2}||\tilde{\mathbf{a}}^-||||\tilde{\mathbf{a}}^+||(\mathbf{b}^- \times \mathbf{b}^+)$$

$$= \frac{||\tilde{\mathbf{a}}^-||||\tilde{\mathbf{a}}^+||}{2||\tilde{\mathbf{b}}^-||||\tilde{\mathbf{b}}^+||}(\tilde{\mathbf{b}}^- \times \tilde{\mathbf{b}}^+) = \frac{||\tilde{\mathbf{a}}^-||||\tilde{\mathbf{a}}^+||}{||\tilde{\mathbf{b}}^-||||\tilde{\mathbf{b}}^+||}(\mathbf{b}_1 \times \mathbf{b}_2)$$

$$\implies (\mathbf{a}_1 \times \mathbf{a}_2) \cdot (\mathbf{b}_1 \times \mathbf{b}_2) > 0$$

Thus, the cross products are aligned in this configuration, and from the lemma in the general case proof, the optimal rotation is a rotation about the z-axis.

From the dot product equality above, we can deduce that $\mathbf{a}^+$ is equidistant from $\mathbf{a}_1, \mathbf{a}_2$. The dot product calculates the cosine of the angle between linearly independent unit vectors measured in the plane spanned by the vectors ($z = 0$ in our case). We know from the proof in the general case that the dot product of a unit vector in the plane $z = 0$ with itself after a rotation about the z-axis is the cosine of the angle of rotation. That angle is measured in the plane perpendicular to the axis of rotation, which is also the plane $z = 0$. Thus, constructing rotations $\mathbf{R}_{\mathbf{a}_1}$ and $\mathbf{R}_{\mathbf{a}_2}$ which rotate $\mathbf{a}^+$ about the z-axis to $\mathbf{a}_1$ and $\mathbf{a}_2$ respectively, we can write the following:

$$\mathbf{a}_1 \cdot \mathbf{a}^+ = \mathbf{a}_2 \cdot \mathbf{a}^+ = \mathbf{a}^+ \cdot (\mathbf{R}_{\mathbf{a}_1}\mathbf{a}^+) = \mathbf{a}^+ \cdot (\mathbf{R}_{\mathbf{a}_2}\mathbf{a}^+) = cos(\phi)$$

where $\phi$ denotes the angle of rotation of $\mathbf{R}_{\mathbf{a}_1}$, making $|\phi|$ (canonically positive) the angle between $\mathbf{a}_1$ and $\mathbf{a}^+$. In general, $\mathbf{R}_{\mathbf{a}_1} \neq \mathbf{R}_{\mathbf{a}_2}$, otherwise $\mathbf{a}_1$ and $\mathbf{a}_2$ would be identical. In order for the above to still hold, the angle of rotation of $\mathbf{R}_{\mathbf{a}_2}$ must have the same magnitude but opposite sign of $\phi$. A similar statement can be made for the target points.

Let $\mathbf{R}_{\mathbf{b}_1}$ and $\mathbf{R}_{\mathbf{b}_2}$ represent rotations about the z-axis that align $\mathbf{b}^+$ with $\mathbf{b}_1$ and $\mathbf{b}_2$ respectively. Recall $\mathbf{a}^+ = \mathbf{b}^+$. We construct the rotations $\mathbf{R}_1 = \mathbf{R}_{\mathbf{b}_1}\mathbf{R}_{\mathbf{a}_1}^T$ and $\mathbf{R}_2 = \mathbf{R}_{\mathbf{b}_2}\mathbf{R}_{\mathbf{a}_2}^T$ which are also about the z-axis to align $\mathbf{a}_1$ with $\mathbf{b}_1$ and $\mathbf{a}_2$ with $\mathbf{b}_2$ respectively. If $\psi$ is the rotation angle of $\mathbf{R}_{\mathbf{b}_1}$, then the angle of rotation for $\mathbf{R}_1$ is $-\phi + \psi$ since $\mathbf{R}_{\mathbf{a}_1}$ and $\mathbf{R}_{\mathbf{b}_1}$ share the same axis of rotation and transposing a rotation matrix negates the rotation angle. For $\mathbf{R}_2$, the rotation angle is $\phi - \psi$, as $\mathbf{R}_{\mathbf{a}_2}$ rotates by $-\phi$ and $\mathbf{R}_{\mathbf{b}_2}$ by $-\psi$. Thus, the rotation angles of $\mathbf{R}_1$ and $\mathbf{R}_2$ have equal magnitudes but opposite signs.

From the proof in the general case, the optimal rotation $\mathbf{R}$ is the weighted average in the Frobenius sense between the rotations $\mathbf{R}_1$ and $\mathbf{R}_2$ recently constructed. The weighted average rotation maximizes the quantity $Tr(\mathbf{R}\mathbf{B}'^T)$ where $\mathbf{B}' = \sum_i w_i\mathbf{R}_i$ Markley et al. (2007). Given the previously made statements and the fact that $w_1 = w_2$, we can calculate $\mathbf{B}'$ as:

$$\mathbf{R}_1 = \begin{bmatrix} cos(-\phi+\psi) & -sin(-\phi+\psi) & 0 \\ sin(-\phi+\psi) & cos(-\phi+\psi) & 0 \\ 0 & 0 & 1 \end{bmatrix}, \quad \mathbf{R}_2 = \begin{bmatrix} cos(\phi-\psi) & -sin(\phi-\psi) & 0 \\ sin(\phi-\psi) & cos(\phi-\psi) & 0 \\ 0 & 0 & 1 \end{bmatrix},$$

$$\mathbf{B}' = w_1\mathbf{R}_1 + w_2\mathbf{R}_2 = 2w_1 \begin{bmatrix} cos(-\phi+\psi) & 0 & 0 \\ 0 & cos(-\phi+\psi) & 0 \\ 0 & 0 & 1 \end{bmatrix}$$

due to the fact that sine is an odd function and cosine is an even function. Since $\mathbf{R}$ is a rotation about the z-axis, we can directly compute $Tr(\mathbf{R}\mathbf{B}'^T)$ as $2w_1(2cos(-\phi+\psi)cos(\theta)+1)$ where $\theta$ is $\mathbf{R}$'s angle of rotation. We can trivially see that $\theta$ must take on a value of 0 or $\pi$ (mod $2\pi$) to be optimal, depending on the sign of $cos(-\phi+\psi)$ as $w_1$ is positive. That sign can be determined considering $\mathbf{a}^-$ and $\mathbf{b}^-$ are aligned:

$$\tilde{\mathbf{a}}^- \cdot \tilde{\mathbf{b}}^- > 0$$

$$(\mathbf{R}_{\mathbf{a}_1}\mathbf{a}^+ - \mathbf{R}_{\mathbf{a}_2}\mathbf{a}^+) \cdot (\mathbf{R}_{\mathbf{b}_1}\mathbf{b}^+ - \mathbf{R}_{\mathbf{b}_2}\mathbf{b}^+) > 0$$

$$\mathbf{a}^+ \cdot ((\mathbf{R}_{\mathbf{a}_1} - \mathbf{R}_{\mathbf{a}_2})^T(\mathbf{R}_{\mathbf{b}_1} - \mathbf{R}_{\mathbf{b}_2})\mathbf{a}^+) > 0$$

$$cos(-\phi+\psi) - cos(-\phi-\psi) - cos(\phi+\psi) + cos(\phi-\psi) > 0$$

$$2cos(-\phi+\psi) - 2cos(\phi+\psi) > 0$$

Since $\mathbf{a}^+$ and $\mathbf{b}^+$ are also aligned, we can similarly derive $2cos(-\phi+\psi)+2cos(\phi+\psi) > 0$. Adding both inequalities together (valid since they are positive quantities), we find that $cos(-\phi + \psi) > 0$. Thus, $\theta$ must be 0 to maximize $Tr(\mathbf{R}\mathbf{B}'^T)$, resulting in $\mathbf{R}$ being the identity matrix and indicating that the current alignment is the optimal one.

To generalize this, we again apply arbitrary rotations $\mathbf{R}_a, \mathbf{R}_b$ to the reference and target sets respectively, transforming them into $\mathbf{a}'_i, \mathbf{b}'_i$. From the proof in the general case, the new optimal rotation $\mathbf{R}' = \mathbf{R}_b\mathbf{R}\mathbf{R}_a^T = \mathbf{R}_b\mathbf{R}_a^T$. Now, we simply verify below that this rotation aligns $\mathbf{a}'^+$ to $\mathbf{b}'^+$ and $\mathbf{a}'^-$ to $\mathbf{b}'^-$ (combined $\pm$ notation for convenience):

$$\mathbf{a}^\pm = \mathbf{b}^\pm = \frac{\mathbf{a}_1 \pm \mathbf{a}_2}{||\mathbf{a}_1 \pm \mathbf{a}_2||} = \frac{\mathbf{b}_1 \pm \mathbf{b}_2}{||\mathbf{b}_1 \pm \mathbf{b}_2||}$$

$$\frac{\mathbf{R}_b(\mathbf{a}_1 \pm \mathbf{a}_2)}{||\mathbf{a}_1 \pm \mathbf{a}_2||} = \frac{\mathbf{b}'_1 \pm \mathbf{b}'_2}{||\mathbf{b}'_1 \pm \mathbf{b}'_2||}$$

$$\frac{\mathbf{R}_b\mathbf{R}_a^T(\mathbf{a}'_1 \pm \mathbf{a}'_2)}{||\mathbf{a}'_1 \pm \mathbf{a}'_2||} = \frac{\mathbf{b}'_1 \pm \mathbf{b}'_2}{||\mathbf{b}'_1 \pm \mathbf{b}'_2||}$$

$$\mathbf{R}'\mathbf{a}'^\pm = \mathbf{b}'^\pm$$

Since $\mathbf{a}'^+$ and $\mathbf{a}'^-$ are orthogonal, they are also linearly independent, and their transformation uniquely defines the rotation $\mathbf{R}'$, thereby completing the proof.

### B.4.3 AVERAGE OF TWO UNNORMALIZED QUATERNIONS

In Markley et al. (2007), it was shown that the average rotation matrix in the Frobenius sense can be calculated via the quaternion $\mathbf{q}$ which optimizes the following:

$$\mathbf{M} = \sum_i w_i\mathbf{q}_i\mathbf{q}_i^T$$

$$\max_{\mathbf{q}} \mathbf{q}^T\mathbf{M}\mathbf{q} \ s.t. \ ||\mathbf{q}|| = 1$$

Where $\mathbf{q}_i$ are the unit norm quaternions corresponding to the rotations being averaged (sign of $\mathbf{q}_i$ is irrelevant). The solution is the eigenvector corresponding to the largest eigenvalue of $\mathbf{M}$. In the two point approach to Wahba's problem proposed previously, we need to construct two quaternion rotations and average them. The formulation above assumes all quaternions have unit norm. However, it would be computationally advantageous (see Table 2) if we did not have to normalize the constructed rotations, thereby avoiding two square root and division operations. From Markley et al. (2007), it is known that the average rotation in the two rotation case is simply a linear combination of the rotations being averaged. To average unnormalized quaterions $\tilde{\mathbf{q}}_1$ and $\tilde{\mathbf{q}}_2$, we can express $\mathbf{M}$ and $\mathbf{q}$ as:

$$\mathbf{M} = w_1\frac{||\tilde{\mathbf{q}}_2||^2}{||\tilde{\mathbf{q}}_1||^2}\tilde{\mathbf{q}}_1\tilde{\mathbf{q}}_1^T + w_2\tilde{\mathbf{q}}_2\tilde{\mathbf{q}}_2^T$$

$$\mathbf{q} = \mu\tilde{\mathbf{q}}_1 + \nu\tilde{\mathbf{q}}_2$$

where $\mu, \nu$ are scalars. The above takes advantage of the fact that scaling $\mathbf{M}$ does not change its eigenvectors. Thus, we reduce the problem from estimating a unit quaternion to estimating two scalars. As a result, we can rewrite the objective as:

$$\mathbf{\Gamma} = \begin{bmatrix} ||\tilde{\mathbf{q}}_1||^2 & \tilde{\mathbf{q}}_1 \cdot \tilde{\mathbf{q}}_2 \\ \tilde{\mathbf{q}}_1 \cdot \tilde{\mathbf{q}}_2 & ||\tilde{\mathbf{q}}_2||^2 \end{bmatrix}, \ \mathbf{v} = \begin{bmatrix} \mu \\ \nu \end{bmatrix}$$

$$\mathbf{\Lambda}_{1,1} = w_1||\tilde{\mathbf{q}}_1||^2||\tilde{\mathbf{q}}_2||^2 + w_2(\tilde{\mathbf{q}}_1 \cdot \tilde{\mathbf{q}}_2)^2$$

$$\mathbf{\Lambda}_{1,2} = \mathbf{\Lambda}_{2,1} = (w_1 + w_2)||\tilde{\mathbf{q}}_2||^2(\tilde{\mathbf{q}}_1 \cdot \tilde{\mathbf{q}}_2)$$

$$\mathbf{\Lambda}_{2,2} = ||\tilde{\mathbf{q}}_2||^2\left(w_2||\tilde{\mathbf{q}}_2||^2 + \frac{w_1(\tilde{\mathbf{q}}_1 \cdot \tilde{\mathbf{q}}_2)^2}{||\tilde{\mathbf{q}}_1||^2}\right)$$

$$\max_{\mathbf{v}} \mathbf{v}^T\mathbf{\Lambda}\mathbf{v} \ s.t. \ \mathbf{v}^T\mathbf{\Gamma}\mathbf{v} = 1$$

where $\cdot$ denotes the usual vector dot product. $\mathbf{\Gamma}$ is the quadratic constraint ensuring that the linear combination of $\tilde{\mathbf{q}}_1$ and $\tilde{\mathbf{q}}_2$ has unit norm, and $\mathbf{\Lambda}$ is the new 2x2 objective to optimize over. Using

the method of Lagrange multipliers, we find that the solution to the above takes the form of a generalized eigenvalue problem $\mathbf{\Lambda}\mathbf{v} = \lambda\mathbf{\Gamma}\mathbf{v}$. Note that the scaling constraint $\mathbf{\Gamma}$ is positive semidefinite, generally representing the equation of an ellipse. Assuming $\mathbf{\Gamma}$ is invertible and well-conditioned (it is discussed later when this is not the case), the solution is the eigenvector of $\mathbf{\Gamma}^{-1}\mathbf{\Lambda}$ corresponding to the largest eigenvalue. Through simplification and scaling, we can express the matrix similarly as:

$$\mathbf{\Gamma}^{-1}\mathbf{\Lambda} \sim \begin{bmatrix} w_1||\tilde{\mathbf{q}}_1||^2||\tilde{\mathbf{q}}_2||^2 & w_1||\tilde{\mathbf{q}}_2||^2(\tilde{\mathbf{q}}_1 \cdot \tilde{\mathbf{q}}_2) \\ w_2||\tilde{\mathbf{q}}_1||^2(\tilde{\mathbf{q}}_1 \cdot \tilde{\mathbf{q}}_2) & w_2||\tilde{\mathbf{q}}_1||^2||\tilde{\mathbf{q}}_2||^2 \end{bmatrix}$$

which maintains its eigenvectors from before. Since the matrix is only 2x2, the eigenvector $\mathbf{v}$ corresponding to the largest eigenvalue can be expressed in closed form. Scaling the eigenvector by the constraint $\mathbf{v}^T\mathbf{\Gamma}\mathbf{v} = 1$ and substituting it back into the original linear combination of $\tilde{\mathbf{q}}_1$ and $\tilde{\mathbf{q}}_2$, we obtain the average quaternion as:

$$\mathbf{q} = \frac{\mu\tilde{\mathbf{q}}_1 + \nu\tilde{\mathbf{q}}_2}{\sqrt{||\tilde{\mathbf{q}}_1||^2\mu^2 + ||\tilde{\mathbf{q}}_2||^2\nu^2 + 2(\tilde{\mathbf{q}}_1 \cdot \tilde{\mathbf{q}}_2)\mu\nu}}$$

where the values $\mu$ and $\nu$ can be expressed equivalently in two ways:

$$\tau^{(1)} = (w_1 - w_2)||\tilde{\mathbf{q}}_1||^2||\tilde{\mathbf{q}}_2||^2, \ \ \omega^{(1)} = 2w_1||\tilde{\mathbf{q}}_2||^2(\tilde{\mathbf{q}}_1 \cdot \tilde{\mathbf{q}}_2), \ \ \nu^{(1)} = 2w_2||\tilde{\mathbf{q}}_1||^2(\tilde{\mathbf{q}}_1 \cdot \tilde{\mathbf{q}}_2)$$

$$\mu^{(1)} = \tau^{(1)} + \sqrt{(\tau^{(1)})^2 + \omega^{(1)}\nu^{(1)}}$$

or

$$\tau^{(2)} = (w_2 - w_1)||\tilde{\mathbf{q}}_1||^2||\tilde{\mathbf{q}}_2||^2, \ \ \omega^{(2)} = 2w_2||\tilde{\mathbf{q}}_1||^2(\tilde{\mathbf{q}}_1 \cdot \tilde{\mathbf{q}}_2), \ \ \mu^{(2)} = 2w_1||\tilde{\mathbf{q}}_2||^2(\tilde{\mathbf{q}}_1 \cdot \tilde{\mathbf{q}}_2)$$

$$\nu^{(2)} = \tau^{(2)} + \sqrt{(\tau^{(2)})^2 + \omega^{(2)}\mu^{(2)}}$$

Both yield the same result except when $\tilde{\mathbf{q}}_1 \cdot \tilde{\mathbf{q}}_2 = 0$ in which case the rotation corresponding to the larger weight is chosen. If $w_1 = w_2$ in that case, then there is no unique solution and either of the rotations can be selected. The former solution set is used when $w_1 > w_2$ and the latter is used when $w_1 \leq w_2$ as to approach the correct value as $\tilde{\mathbf{q}}_1 \cdot \tilde{\mathbf{q}}_2 \to 0$.

Note that the denominator in the expression for the average quaternion is simply $\sqrt{\mathbf{v}^T\mathbf{\Gamma}\mathbf{v}}$. Previously, $\mathbf{\Gamma}$ was assumed non-singular and well-conditioned, but there are two cases in practice where this fails to hold. The first is when $\tilde{\mathbf{q}}_1$ and $\tilde{\mathbf{q}}_2$ are linearly dependent, i.e. they represent the same rotation. If we choose the solution constants above by the previously described strategy and examine the expressions for $\mu$ and $\nu$, then it can be seen that $\mathbf{v}^T\mathbf{\Gamma}\mathbf{v}$ is in fact strictly positive for nontrivial solutions $\mathbf{v}$ and nonzero weights/magnitudes. Furthermore, it can also be seen that $\mu\tilde{\mathbf{q}}_1$ and $\nu\tilde{\mathbf{q}}_2$ share the same direction in this case and thus cannot cancel out. The second case occurs when the magnitudes of $\tilde{\mathbf{q}}_1$ and/or $\tilde{\mathbf{q}}_2$ are small, causing $\mathbf{\Gamma}$ to be ill-conditioned. This case can be avoided by using the strategy described in Appendix D.2 to only obtain quaternions of sufficient magnitude or by simply scaling/normalizing the rotations when necessary.

### B.4.4 DEGENERATE CASE SOLUTION

The degenerate case occurs when either of the cross products of the reference or target points vanish, and the previous approaches for the two point case cannot be applied. This is because the solution is no longer unique. A particular one can be efficiently found through the following approach.

We assume without loss of generality that the target points are collinear (the reference points may or may not be) and the first target point is aligned with the x-axis (i.e. $\mathbf{b}_1 = (1, 0, 0)$). In this case, the last two columns of the constraint $\mathbf{C}_i$ (Eq. (17)) vanish. We can thus write our optimization as:

$$\mathbf{C}_i = \begin{bmatrix} (m - x)i & y - zi \\ -y - zi & (x + m)i \end{bmatrix}, \ \mathbf{u} = \begin{bmatrix} \alpha \\ \beta \end{bmatrix}$$

$$\mathbf{Z} = \sum_i w_i\mathbf{C}_i^H\mathbf{C}_i$$

$$\min_{\mathbf{u}} \mathbf{u}^H\mathbf{Z}\mathbf{u} \ s.t. \ \mathbf{u}^H\mathbf{u} = 1$$

This optimization is simpler than before and can now be solved directly over the special unitary parameters. Since $\mathbf{Z}$ is Hermitian and positive semidefinite, the solution is the complex eigenvector of $\mathbf{Z}$ corresponding to the smallest eigenvalue. For reference points $\mathbf{a}_i = (x_i, y_i, z_i)$, this can be expressed in closed form as:

$$\tilde{\mathbf{u}} = \begin{bmatrix} w_1 x_1 + w_2 x_2 + ||w_1 \mathbf{a}_1 + w_2 \mathbf{a}_2|| \\ w_1 z_1 + w_2 z_2 - (w_1 y_1 + w_2 y_2)i \end{bmatrix}$$

or

$$\tilde{\mathbf{u}} = \begin{bmatrix} w_1 x_1 - w_2 x_2 + ||w_1 \mathbf{a}_1 - w_2 \mathbf{a}_2|| \\ w_1 z_1 - w_2 z_2 - (w_1 y_1 - w_2 y_2)i \end{bmatrix}$$

where $\tilde{\mathbf{u}}$ is the unnormalized eigenvector and the correct solution depends on the target points' configuration. If the dot product of the target points is positive, then the first expression is correct. Otherwise, the second is correct. Note that eigenvectors are only unique up to scale, so even after normalizing the solution so that $\mathbf{u}^H \mathbf{u} = 1$, we can still apply an arbitary unitary scaling of $e^{\theta i}$. This corresponds to a rotation about the x-axis and parameterizes the family of optimal solutions.

For arbitrary collinear target points, we simply need to find any rotation aligning the x-axis to the first target point $\mathbf{b}_1$ and then compose it with $\mathbf{u}$. If the reference points were collinear instead, we can swap the reference and target points in the above approach and invert the rotation afterwards. In practice, we would choose the more degenerate (i.e. larger dot product magnitude) of the two sets to treat as collinear.

Examining the solution closer, it can be seen that $\mathbf{u}$ represents a rotation aligning a weighted combination of the reference points we refer to as the "weighted average" with the x-axis. The weighted average takes the form of a sum $(w_1 \mathbf{a}_1 + w_2 \mathbf{a}_2)$ or difference $(w_1 \mathbf{a}_1 - w_2 \mathbf{a}_2)$ depending on the sign of the dot product between target points. This suggests that a more straightforward approach in practice would be to simply calculate the normalized weighted average of the reference points and align it with $\mathbf{b}_1$ directly. This generalizes to the case when the reference points are collinear similarly to before. If the weighted average is zero, then any rotation is optimal.

## C  Additional Stereographic Solution Details

### C.1  Recovering R

The solution $\mathbf{U}$ obtained precisely satisfies the relation in Eq. (34). However, using the maps laid out in Eqs. (35) and (36) directly will lead to a rotation $\mathbf{R_U}$ that is not necessarily equivalent to the desired $\mathbf{R}$ in Eq. (1). This is because our choice of $\mathbf{p}^*$ and choice of isomorphism between quaternions and special unitary matrices can each add an implicit orthogonal transformation in their map. Since their combined transformation $\boldsymbol{\Psi}$ and its inverse are applied before and after estimation respectively, the relationship between $\mathbf{U}$ and $\mathbf{R}$ is characterized by the conjugate transformation:

$$\mathbf{R} = \boldsymbol{\Psi}^T \mathbf{R_U} \boldsymbol{\Psi} \tag{44}$$

For our definitions, we find that $\boldsymbol{\Psi}$ is simply a 90 degree rotation about the y-axis. When applied directly to the resulting $\mathbf{q}$ from the algorithm, the transformed quaternion is given as:

$$\mathbf{q}^* = w_q - z_q i + y_q j + x_q k \tag{45}$$

which is just a permutation/negation of the elements of $\mathbf{q}$. We can verify that mapping $\mathbf{q}^*$ to $\mathbf{R}$ via Eq. (36) indeed gives us the true optimal solution to the problem.

### C.2  General Stereographic Constraint

The generalized constraint between complex rays $[z_1, z_2]^T$ and $[p_1, p_2]^T$ where $z_1 = x_1 + y_1 i$, $z_2 = x_2 + y_2 i$, $p_1 = m_1 + n_1 i$, and $p_2 = m_2 + n_2 i$ is given by:

$$w_i' = \frac{4w_i}{(|z_1|^2 + |z_2|^2)(|p_1|^2 + |p_2|^2)}$$

$$\mathbf{A}_i \mathbf{u} = \begin{bmatrix} -z_1 p_2 & -z_2 p_2 & p_1 z_2 & -p_1 z_1 \end{bmatrix} \mathbf{u} = 0$$

for complex inputs and below for real inputs:

$$\mathbf{D}_{i,0} = \begin{bmatrix} m_2x_1 - m_1x_2 + n_1y_1 - n_2y_1 & -m_2y_1 - m_1y_2 - n_2x_1 - n_1x_2 \\ m_2y_1 - m_1y_2 + n_2x_1 - n_1x_2 & m_2x_1 + m_1x_2 - n_1y_2 - n_2y_1 \end{bmatrix}$$

$$\mathbf{D}_{i,1} = \begin{bmatrix} m_1x_1 + m_2x_2 - n_1y_1 - n_2y_2 & m_1y_1 - m_2y_2 + n_1x_1 - n_2x_2 \\ m_1y_1 + m_2y_2 + n_1x_1 + n_2x_2 & m_2x_2 - m_1x_1 + n_1y_1 - n_2y_2 \end{bmatrix}$$

$$\mathbf{D}_i\mathbf{q} = [\mathbf{D}_{i,0} \quad \mathbf{D}_{i,1}]\, \mathbf{q} = 0$$

We can verify that with $z_2 = 1$ and $p_2 = 1$, we obtain the original results in Eq. (9) and Eq. (11). Furthermore, we can use $z_2 = 0$ and $p_2 = 0$ to calculate results involving the projective point at infinity. Thus, there are no singularities using the general constraint. From this, we can derive similar formulas and algorithms for the one and two point cases as those proposed earlier.

Similarly, the following is the general constraint for estimating a Möbius transformation from stereographic inputs:

$$\mathbf{A}'_i\mathbf{m} = [-z_1p_2 \quad -z_2p_2 \quad p_1z_1 \quad p_1z_2]\,\mathbf{m} = 0$$

## D  ROTATIONS OF EXACT ALIGNMENT

The equations in this section are derived from the constraint in Eq. (18) for 3D points. However, we can easily derive similar equations for stereographic points using Eq. (11).

### D.1  ONE-POINT CASE

Finding a rotation that aligns two unit vectors (i.e. $\mathbf{b} = \mathbf{R}\mathbf{a}$) is a special case of Wahba's problem where $n = 1$. Since aligning a pair of points constrains two out of three rotational degrees of freedom ($\mathbf{D}_i$ and $\mathbf{Q}_i$ have rank 2), there are infinite solutions in this case. The rotation whose axis is the cross product of the points is often chosen for geometric simplicity and can be calculated efficiently as:

$$s = \sqrt{2(1 + \mathbf{a} \cdot \mathbf{b})}$$

$$\mathbf{q} = (\frac{s}{2}, \frac{\mathbf{a} \times \mathbf{b}}{s}) \tag{46}$$

Instead, we may choose another convention where we constrain an element of the quaternion to be 0. Since the points can be perfectly aligned, $\mathbf{q}^T\mathbf{G}_S\mathbf{q} = 0$, so $\mathbf{q} \in Null(\mathbf{Q}_i)$. Leveraging this fact, we can simply take two linearly independent rows from $\mathbf{Q}_i$ and set them to 0 explicitly, imposing a rank 2 constraint. Given the homogeneous nature of this system, we can disregard the weight and determine the rotation using straightforward linear algebra techniques. Each row below is a member of the kernel that has a quaternion element equal to 0 (note only two rows are linearly independent):

$$\begin{Bmatrix} 0 & x+m & y+n & z+p \\ x+m & 0 & z-p & n-y \\ y+n & p-z & 0 & x-m \\ z+p & y-n & m-x & 0 \end{Bmatrix} \in ker(\mathbf{Q}_i) \tag{47}$$

Normalizing any nonzero row of Eq. (47) gives an optimal rotation. Compared to Eq. (46), this approach has several advantages. First, the rotation is simpler to construct. Second, one of its elements is guaranteed to be 0, so composing rotations and rotating points requires fewer operations and memory accesses. This is particularly true for the first row of Eq. (47) as it represents a 180 degree rotation whose action on a point can be more efficiently computed as a reflection about an axis. Finally, Eq. (46) has a singularity when the cross product vanishes. Although each row of Eq. (47) has its own singular region, it is straightforward to systematically select another row that is well-defined in that region.

### D.2  NOISELESS TWO-POINT CASE

With two independent sets of correspondences, we are able to fully constrain the rotation to a unique one. If we assume that the two sets can be aligned perfectly, then we can recover an optimal rotation

from the intersection of the constraint kernels. Two independent rows of Eq. (47) can be basis vectors for the kernel of $\mathbf{Q}_1$. We can determine the optimal rotation by finding the member of $ker(\mathbf{Q}_1)$ (represented as a linear combination of basis vectors) that is orthogonal to an independent row of $\mathbf{Q}_2$. For example, with the last two rows of Eq. (47) as a basis of $\mathbf{Q}_1$ and the first row of $\mathbf{Q}_2$, we can solve for the linear combination weights $a, b$ (note scale is arbitrary):

$$\begin{bmatrix} 0 \\ x_2 - m_2 \\ y_2 - n_2 \\ z_2 - p_2 \end{bmatrix} \cdot (a \begin{bmatrix} z_1 + p_1 \\ y_1 - n_1 \\ m_1 - x_1 \\ 0 \end{bmatrix} + b \begin{bmatrix} y_1 + n_1 \\ p_1 - z_1 \\ 0 \\ x_1 - m_1 \end{bmatrix}) = 0$$

$$a = (x_2 - m_2)(z_1 - p_1) + (z_2 - p_2)(m_1 - x_1)$$

$$b = (x_2 - m_2)(y_1 - n_1) + (y_2 - n_2)(m_1 - x_1)$$

Substituting $a$ and $b$ back into the linear combination and dividing by $m_1 - x_1$ gives the result from Eq. (20): This result is equivalent to the simple estimators found in Markley (1999); Choukroun (2009). However, an issue with this approach is that the singular region of this estimator is not simple, and the equation fails to produce a valid rotation under several conditions (see Peng and Choukroun (2024)). Rather than checking each condition with a threshold or applying sequential rotations to avoid these cases like other kernel methods, we can more systematically select the three vectors in our computation to guarantee a valid result.

In general, we observe that for a point pair, either $\mathbf{a} + \mathbf{b}$ or $\mathbf{a} - \mathbf{b}$ will have at least one significantly nonzero element. We can select the two rows from Eq. (47) corresponding to a nonzero element from these vectors for the first point pair to ensure linearly independent kernel vectors. We then choose one of the two rows of $\mathbf{Q}_2$ corresponding to a nonzero element of $\mathbf{a} + \mathbf{b}$ or $\mathbf{a} - \mathbf{b}$ for the second point pair to solve for the rotation. For instance, if $x_1 + m_1 \neq 0$ and $y_2 + n_2 \neq 0$, we can choose the first two rows of Eq. (47) and the last row of $\mathbf{Q}_2$ to produce another equation for the rotation:

$$\mathbf{k}_1 = \begin{bmatrix} p_1 - z_1 & -y_1 - n_1 & x_1 + m_1 \end{bmatrix}^T$$

$$\mathbf{k}_2 = \begin{bmatrix} z_1 + p_1 & y_1 - n_1 & m_1 - x_1 \end{bmatrix}^T$$

$$\mathbf{k}_3 = \begin{bmatrix} p_2 - z_2 & -y_2 - n_2 & x_2 + m_2 \end{bmatrix}^T$$

$$\tilde{\mathbf{q}} = \begin{bmatrix} \mathbf{k}_1 \times \mathbf{k}_3 \\ \mathbf{k}_2 \cdot \mathbf{k}_3 \end{bmatrix} \tag{48}$$

Though the dot and cross products are in different indices from before, the formulation is equally simple to compute. We select the nonzero elements by largest magnitude for robustness. At least one of the two rows we select from $\mathbf{Q}_2$ will yield a valid rotation for $\mathbf{a}_1 \times \mathbf{a}_2 \neq 0$. Otherwise, the rotation is any kernel vector of $\mathbf{Q}_1$. We verify row validity by checking if either coefficient $a$ or $b$ for the relevant constraints is nonzero. Those coefficients are always reused in the final rotation calculation (e.g. $a$ and $b$ are the second and first elements respectively in Eq. (48)). This process therefore covers the whole domain and only requires a handful of operations and comparisons even in the worst case.

## E  BACKPROPAGATION DERIVATIVES

For a simple complex square matrix $\mathbf{G}$, the derivative of an eigenvector $\mathbf{v}$ of $\mathbf{G}$ with respect to the elements of $\mathbf{G}$ can be computed as Magnus (1985):

$$d\mathbf{v} = (\lambda\mathbf{I} - \mathbf{G})^+ (\mathbf{I} - \frac{\mathbf{v}\mathbf{v}^H}{\mathbf{v}^H\mathbf{v}})(d\mathbf{G})\mathbf{v}$$

where $\lambda$ is the eigenvalue corresponding to $\mathbf{v}$, $\mathbf{I}$ is the identity matrix, and $^+$ denotes the Moore-Penrose pseudoinverse. Typically, $\mathbf{v}^H\mathbf{v} = 1$ by convention for most eigenvector solvers. In our original problem (Eq. (14)), $\mathbf{G}_M$ is Hermitian as opposed to a general matrix, so the elements of $\Theta$ are repeated in the matrix through conjugation. Using complex differentiation conventions consistent with many deep learning frameworks, the loss derivative can be written as:

$$\frac{d\mathcal{L}}{d(\mathbf{G}_M)_{i,j}} = \frac{1}{2}\left(\left\langle \frac{d\mathbf{v}}{d\mathbf{G}_{i,j}}, \frac{d\mathcal{L}}{d\mathbf{v}} \right\rangle + \left\langle \frac{d\mathcal{L}}{d\mathbf{v}}, \frac{d\mathbf{v}}{d\mathbf{G}_{j,i}} \right\rangle\right)$$

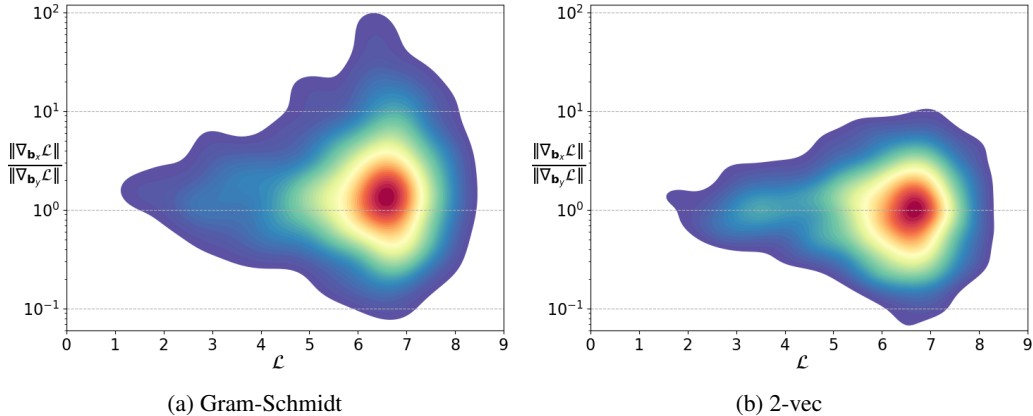

Figure 4: Density plot of loss gradient ratios for Gram-Schmidt and 2-vec. The x-axis represents the loss $\mathcal{L}$, and the y-axis shows the ratio of loss gradient magnitudes $\|\nabla_{\mathbf{b}_x}\mathcal{L}\|/\|\nabla_{\mathbf{b}_y}\mathcal{L}\|$ for the predicted rotation axes $\mathbf{b}_x$ and $\mathbf{b}_y$. See Appendix F for details. 2-vec exhibits noticeably lower variance, suggesting more stable gradients during learning.

where $\langle\cdot,\cdot\rangle$ denotes the complex inner product and $\mathcal{L}$ is the scalar loss. $\frac{d\mathcal{L}}{d\Theta}$ can be extracted from the upper triangular portion of $\frac{d\mathcal{L}}{d\mathbf{G}_M}$ (after reshaping to 4 x 4), multiplying by 2 for the off-diagonal parameters to include the lower portion contribution. This method avoids the need for the other eigenvectors or eigenvalues of $\mathbf{G}_M$ that weren't used in the forward pass.

For QuadMobiusSVD (Eq. (25)), the backpropagation must go through the SVD operation $\mathbf{M} = \mathbf{U}\Sigma\mathbf{V}^H$. It is well known that the nearest unitary matrix corresponds to the unitary component $\mathbf{Q}$ of the polar decomposition of $\mathbf{M} = \mathbf{Q}\mathbf{P}$, where $\mathbf{P}$ is a positive semidefinite and Hermitian matrix Keller (1975). Thus, instead of backpropagating through the SVD components individually, we can backpropagate through $\mathbf{Q}$ in a more direct manner. Appendix B.3.2 outlines the details of the derivative of $\mathbf{Q}$ with respect to the elements of $\mathbf{M}$. Given the well-known relationships between the polar decomposition and SVD ($\mathbf{Q} = \mathbf{U}\mathbf{V}^H$, $\mathbf{P} = \mathbf{V}\Sigma\mathbf{V}^H$), we can reuse the SVD elements from the forward pass to calculate the derivative more simply as:

$$\mathbf{S} = diag(\Sigma) \oplus diag(\Sigma)$$

$$d\mathbf{Q} = \mathbf{U}\Big(\frac{\mathbf{U}^H(d\mathbf{M})\mathbf{V} - \mathbf{V}^H(d\mathbf{M}^H)\mathbf{U}}{\mathbf{S}}\Big)\mathbf{V}^H$$

where $\oplus$ denotes an outer sum operation, and the division is Hadamard division (element-wise). From this equation, the numerical complex derivative can be expressed as follows (note the indices, $\mathbf{F}$ is 2 x 2 x 2 x 2):

$$\mathbf{F}_{j,m,l,k} = \mathbf{U}_{j,k}(\mathbf{V}^H)_{l,m}$$

$$\frac{d\mathcal{L}}{d\mathbf{M}_{j,m}} = \Big\langle\mathbf{U}\Big(\frac{\mathbf{F}^H_{j,m}}{\mathbf{S}}\Big)\mathbf{V}^H, \frac{d\mathcal{L}}{d\mathbf{Q}}\Big\rangle_F - \Big\langle\frac{d\mathcal{L}}{d\mathbf{Q}}, \mathbf{U}\Big(\frac{\mathbf{F}_{j,m}}{\mathbf{S}}\Big)\mathbf{V}^H\Big\rangle_F$$

where $\langle\cdot,\cdot\rangle_F$ denotes the complex Frobenius inner product.

The remaining operations in the maps are algebraically straightforward to differentiate through. We observe that the previous formulas compute the same gradients as PyTorch's automatic differentiation through complex functions `torch.linalg.eigh` and `torch.linalg.svd` but in a more streamlined manner.

## F  THEORETICAL INVESTIGATIONS OF REPRESENTATIONS

**2-vec**  The core idea behind 2-vec lies in leveraging a more optimal projection (in the sense of Wahba's problem) than Gram-Schmidt to improve learning performance without increasing computational cost or dimensionality. To theoretically support this, we replicate the gradient analysis

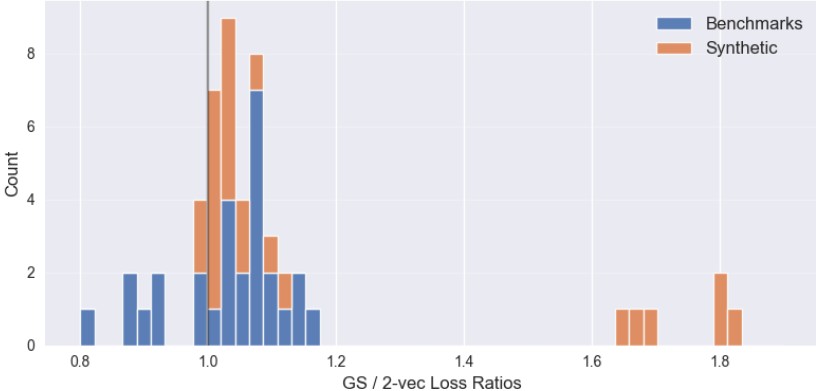

Figure 5: Histogram of loss ratio between Gram-Schmidt (GS) and 2-vec representations for all reported figures in this paper (accuracy converted to 1-Acc to maintain directionality). Gram-Schmidt performs around 11% worse on average than 2-vec, with some experiments showing a large discrepancy in performance between the two. 2-vec performed better on 41 out of 52 reported metrics.

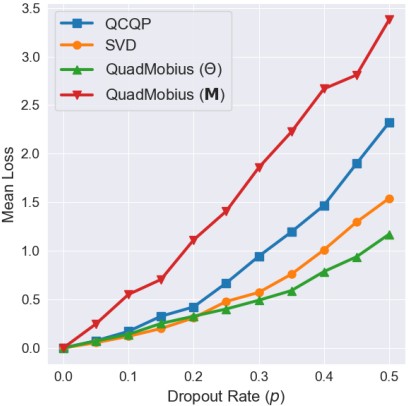

Figure 6: Plot of mean loss (Chordal L2) against dropout rate of map representations. $\Theta$ and $\mathbf{M}$ denote whether dropout was applied to map inputs or intermediate representation for QuadMobius.

experiment from Geist et al. (2024) which evaluates how learning signals propagate through the representations. We first generate a thousand random 6D vectors, each with components sampled uniformly from [-2, 2]. Each vector is split into two 3D components, $\mathbf{b}_x$ and $\mathbf{b}_y$, representing predicted target $x, y$ coordinate axes. These are then mapped to a rotation matrix using both the Gram-Schmidt and 2-vec methods. For each mapping, we compute the Frobenius norm loss $\mathcal{L}$ between the resulting rotation and the identity matrix. We then calculate the gradient magnitudes of $\mathcal{L}$ with respect to $\mathbf{b}_x$ and $\mathbf{b}_y$ and analyze their ratio. The results are plotted in Fig. 4. We can see that the gradient ratios for 2-vec are more tightly concentrated around 1, indicating a relatively balanced gradient flow between the two vectors. In contrast, the Gram-Schmidt method exhibits a wider distribution with significant skew, often yielding ratios in the range of 10–100 which highlights its disproportionate focus on $\mathbf{b}_x$. These results support the hypothesis that 2-vec facilitates more stable gradients for optimization. The empirical performance gap between the two methods is visualized in Fig. 5.

**QuadMobius** In our experiments, QuadMobius has consistently shown strong performance as a learning representation. To better understand why, we conduct some experiments to probe its behavior. We begin by generating one thousand realistic map inputs $\Theta$ for each representation using trained models from a synthetic Wahba's problem (trial #15 in Appendix G.2.1). All models are fed the same noiseless inputs on which they perform equivalently for fair comparison. In the first

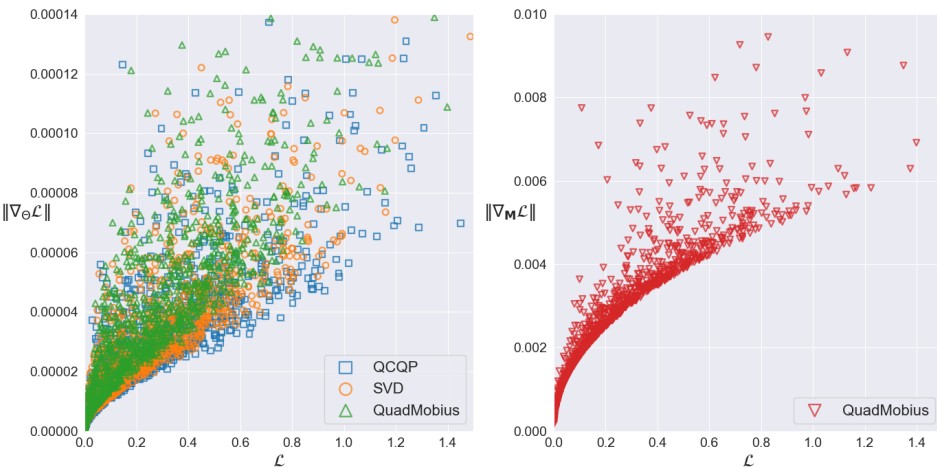

Figure 7: Distribution plot of loss gradient magnitudes against loss $\mathcal{L}$ (Chordal L2). The left shows the gradient with respect to the map inputs $\Theta$, while the right shows the gradient with respect to the Möbius transformation $\mathbf{M}$ estimated from eigendecomposition in QuadMobius.

experiment, we test how resilient each map is to corrupted inputs by applying dropout. Fig. 6 shows the results of applying increasing dropout probability to $\Theta$ on mean loss. For QuadMobius, we also test applying dropout to its intermediate Möbius transformation $\mathbf{M}$ instead (real and imaginary parts treated independently). While we might expect the sensitivity to dropout to decrease with dimensionality, this is not necessarily the case as seen with QCQP. Notably, QuadMobius appears to be the most resilient to dropout on $\Theta$, but is also the most sensitive when applied to $\mathbf{M}$. For the second experiment, we replace 10% of the model inputs with outlier points from another rotation, simulating out-of-domain inference. Fig. 7 plots the distribution of loss gradient magnitudes against loss. Gradients with respect to $\Theta$ are broadly similar across all maps, consistent with their relative performance on the task. In contrast, gradients with respect to $\mathbf{M}$ in QuadMobius are both significantly larger and more tightly concentrated, following a square root trend. Together, these two experiments suggest that QuadMobius's eigendecomposition step enables the learning of a stable intermediate representation that is buffered against poor inputs, while its subsequent $SU(2)$ projection ensures predictable, high-fidelity gradient flow, leading to its strong empirical performance.

**$SU(2)$**  A natural question is whether we can just directly predict an $SU(2)$ representation and project it onto the manifold. This approach is simpler than QuadMobius and still provides an overparameterized representation (8D). However, like quaternions, $SU(2)$ suffers from the issue of double cover. Both Möbius transformation predictions $\mathbf{M}$ and $-\mathbf{M}$ map to the same 3D rotation, introducing ambiguity in learning. Double cover also weakens the interpretation that the rows of $\mathbf{M}$ provide independent quaternion estimates (similar to theoretical arguments of information averaging in SVD and QCQP). This is because the projection to $SU(2)$ minimizes the Frobenius norm which is affected by sign inconsistencies across rows. Empirically, projection alone appears less stable for learning as $SU(2)$ prediction performed substantially worse in synthetic experiments than QuadMobius (and only marginally better than quaternion representation).

To further validate the necessity of both steps in the QuadMobius approach, we conducted a toy ablation experiment in Table 5. We took 10k random map inputs and mapped them to quaternions. We then calculate the squared quaternion loss (accounting for sign) against a set of random ground truth quaternions and compare the loss gradient magnitudes of the inputs for the different map variants. The variants include SVD projection only (8D $\to$ $SU(2)$), Eigendecomposition only (16D $\to$ Möbius transformation $\mathbf{M}$, taking the first row of $\mathbf{M}$ as a quaternion with and without normalization), and QuadMobius. The percentiles of the gradient distributions and their subsequent percentile ranges are shown in the table below. The QuadMobius approach yields a significantly tighter distribution and a lower amount of large outlying values than the other isolated components, suggesting that it provides more stable gradients for learning with both eigendecomposition and projection.

| Method | 10% | 25% | 50% | 75% | 90% | 25-75% | 10-90% |
|---|---|---|---|---|---|---|---|
| Projection | 2.04e-5 | 2.66e-5 | 3.23e-5 | 3.65e-5 | 4.02e-5 | 9.90e-6 | 1.98e-5 |
| Eig. (no norm) | 2.06e-5 | 2.87e-5 | 3.41e-5 | 3.88e-5 | 4.08e-5 | 1.00e-5 | 2.02e-5 |
| Eig. (norm) | 1.43e-5 | 2.07e-5 | 2.51e-5 | 3.01e-5 | 3.20e-5 | 9.41e-6 | 1.78e-5 |
| QuadMobius | 1.46e-5 | 1.79e-5 | 2.20e-5 | 2.49e-5 | 2.64e-5 | **6.98e-6** | **1.17e-5** |

Table 5: Toy ablation experiment showing gradient magnitude distributions for isolated components of QuadMobius algorithm. Bold indicates lowest for spread quantities.

# G    Experiments

## G.1    Experiment Settings and Details

These are the specific experiment settings used to obtain the results in our learning experiments.

**ModelNet10-SO3**    ADAM optimizer, learning rate 5e-4, NVIDIA L1 GPU, batch size 100, Chordal L2 loss, 300/400/800 epochs respectively for chair/sofa/toilet to train for roughly equal iterations given dataset size differences. Architecture is ShuffleNetV2-1.5 backbone Ma et al. (2018) (used for its quick training) pretrained on ImageNet weights followed by two fully connected layers featuring ReLU activation and dropout applied before the layers with probability 0.4 and 0.25 respectively. Models saved by best average rotation error.

**Inverse Kinematics**    Original author source code and settings Zhou et al. (2019) were utilized. Trained on NVIDIA L1 GPU for 2 million iterations. Epoch with lowest median rotation error was chosen for results.

**Camera Pose Estimation**    Training code and settings obtained from Chen et al. (2022). Model initialized from pretrained GoogleNet weights recommended by original paper. Used NVIDIA L1 GPU and beta values 500/100/1500 for King's College/Shop Facade/Old Hospital. Trained for 1200 epochs with batch size 75. Models saved every 5 epochs, and models from last 300 epoch were used for testing (batch size 1 in testing). Epoch with lowest median rotation error was chosen for results.

## G.2    Additional Experiments

### G.2.1    Learning Wahba's Problem

To evaluate our rotation representations more robustly across various conditions, we replicate the synthetic learning experiments from Peretroukhin et al. (2020); Levinson et al. (2020); Zhou et al. (2019), using a fully-connected neural network from Peretroukhin et al. (2020) to learn the solution to Wahba's problem. Problem points and rotations are generated according to same procedure described in Section 5.1. Each epoch, we dynamically generate 25,600 training samples and validate on a fixed set of the same size ($\epsilon_{noise} = 0.01$ added to all samples). The models are trained for 1000 epochs with ADAM optimizer on an NVIDIA T4 GPU. In addition to Chordal L2, we also define the loss function Chordal L1 analogously as the sum of absolute differences between the elements of $\mathbf{R}_{pred}$ and $\mathbf{R}_{gt}$. Finally, given our complex representations, we also evaluate training complex-valued networks Liao (2023); Barrachina et al. (2023) of equivalent size for the task with stereographic complex inputs (Eq. (30)). For real-valued representations, we take the real part of the model output in this case.

As expected, the compact representations (Euler, Quat) performed relatively poorly. Overall, the best performers (QCQP, SVD, QuadMobiusAlg, QuadMobiusSVD) were all quite competitive with each other, having similar results and convergence rates. However, the QuadMobius representations together demonstrated an edge, leading most of the epochs and having the lowest error in majority of trials. Although mathematically equivalent, the two approaches produced different results with neither approach consistently outperforming the other. On the other hand, 2-vec outperformed the other non-eigendecomposition representations (including Gram-Schmidt), beating them on most trials, at times by a large margin. Although significant differences for the complex cases were not observed among representations, some of the complex-valued trials featured the highest leader

| # | $n$ | LR | Loss | Dom | Euler | Quat | GS | QCQP | SVD | 2-vec | QMAlg | QMSVD |
|---|---|---|---|---|---|---|---|---|---|---|---|---|
| 1 | 3 | 1e-4 | L2 | ℝ | 9.009/0 | 8.964/1 | 1.761/0 | 1.676/141 | **1.641/696** | 1.701/1 | 1.658/51 | 1.689/110 |
| 2 | 3 | 1e-4 | L2 | ℂ | 119.364/0 | 13.632/0 | 5.768/0 | 4.237/1 | 4.264/1 | 5.781/0 | 3.823/109 | **3.761/889** |
| 3 | 3 | 5e-4 | L2 | ℝ | 12.154/0 | 9.618/0 | 1.583/5 | 1.518/143 | **1.491/582** | 1.560/0 | 1.501/217 | 1.527/53 |
| 4 | 3 | 5e-4 | L2 | ℂ | 119.403/0 | 12.238/0 | 4.016/0 | 3.586/2 | 3.735/6 | 3.917/0 | 3.447/**751** | **3.408**/241 |
| 5 | 3 | 1e-3 | L2 | ℝ | 14.693/0 | 9.159/0 | 1.575/1 | 1.497/170 | 1.509/245 | 1.578/2 | **1.486**/87 | 1.499/**495** |
| 6 | 3 | 1e-3 | L2 | ℂ | 119.397/0 | 11.212/0 | 3.290/24 | 3.289/190 | 3.253/**384** | 3.269/0 | 3.250/110 | **3.232**/292 |
| 7 | 3 | 1e-4 | L1 | ℝ | 8.063/0 | 4.120/0 | 1.603/0 | 1.445/135 | **1.421/622** | 1.570/2 | 1.469/164 | 1.459/77 |
| 8 | 3 | 1e-4 | L1 | ℂ | 119.388/0 | 9.812/0 | 4.734/0 | 3.259/0 | 3.238/1 | 4.663/0 | 2.835/492 | **2.786/507** |
| 9 | 3 | 5e-4 | L1 | ℝ | 8.687/0 | 4.355/0 | 1.459/0 | 1.315/175 | 1.322/279 | 1.416/0 | **1.303/418** | 1.306/128 |
| 10 | 3 | 5e-4 | L1 | ℂ | 119.334/0 | 7.500/0 | 3.290/0 | 2.760/3 | 2.857/3 | 3.113/0 | **2.750/921** | 2.807/73 |
| 11 | 3 | 1e-3 | L1 | ℝ | 10.833/0 | 4.436/0 | 1.434/0 | 1.312/53 | 1.301/**338** | 1.427/0 | 1.317/337 | **1.291**/272 |
| 12 | 3 | 1e-3 | L1 | ℂ | 119.483/0 | 6.930/0 | 2.916/0 | 2.475/92 | **2.447/251** | 2.874/0 | 2.478/211 | 2.472/**446** |
| 13 | 100 | 1e-4 | L2 | ℝ | 3.784/0 | 3.277/0 | 0.569/0 | 0.253/138 | **0.243/389** | 0.313/0 | 0.255/169 | 0.251/304 |
| 14 | 100 | 1e-4 | L2 | ℂ | 48.175/0 | 4.988/0 | 1.400/0 | 0.638/254 | 0.637/136 | 0.850/0 | **0.625**/281 | 0.634/**329** |
| 15 | 100 | 5e-4 | L2 | ℝ | 5.395/0 | 3.712/0 | 0.547/0 | 0.249/121 | 0.247/175 | 0.303/0 | 0.247/**368** | **0.242**/336 |
| 16 | 100 | 5e-4 | L2 | ℂ | 119.370/0 | 5.009/0 | 1.586/0 | **0.831/682** | 0.866/223 | 0.940/0 | 0.866/66 | 0.848/29 |
| 17 | 100 | 1e-3 | L2 | ℝ | 6.608/0 | 3.269/0 | 0.537/0 | **0.243**/292 | 0.272/112 | 0.297/0 | 0.261/**299** | 0.253/297 |
| 18 | 100 | 1e-3 | L2 | ℂ | 118.381/0 | 5.056/0 | 1.480/0 | 0.845/121 | 0.836/**499** | 0.887/0 | 0.859/71 | **0.826**/309 |
| 19 | 100 | 1e-4 | L1 | ℝ | 2.249/0 | 1.794/0 | 0.356/0 | 0.269/293 | **0.261/327** | 0.332/0 | 0.264/130 | 0.265/250 |
| 20 | 100 | 1e-4 | L1 | ℂ | 109.217/0 | 3.209/0 | 0.927/0 | **0.665**/268 | 0.667/**469** | 0.889/0 | 0.669/196 | 0.669/67 |
| 21 | 100 | 5e-4 | L1 | ℝ | 2.666/0 | 1.055/0 | 0.355/0 | 0.275/83 | 0.284/339 | 0.316/1 | 0.289/209 | **0.272/368** |
| 22 | 100 | 5e-4 | L1 | ℂ | 119.299/0 | 1.954/0 | 0.938/0 | 0.883/**780** | 0.877/101 | 0.956/0 | **0.873**/73 | 0.878/46 |
| 23 | 100 | 1e-3 | L1 | ℝ | 3.867/0 | 1.384/0 | 0.366/0 | 0.280/167 | 0.280/316 | 0.331/0 | **0.277/346** | 0.291/171 |
| 24 | 100 | 1e-3 | L1 | ℂ | 83.623/0 | 2.184/0 | 0.952/0 | 0.830/**466** | 0.835/61 | 0.919/0 | **0.826**/366 | 0.849/107 |

Table 6: Trial results for learning Wahba's problem with different rotation representations. $n$ is number of points, LR is learning rate, Loss is type of chordal loss function, Dom is the domain, specifying whether the network is real-valued or complex-valued. Results are shown as $\theta_{err}$/Ldr. pairs where $\theta_{err}$ is average rotation error on validation set, and Ldr. is the number of epochs where that representation was a leader, i.e. had the lowest $\theta_{err}$ overall as of that epoch. Bold indicates best value, underline indicates second best.

counts overall by our representations (e.g. trial #2, trial #10). The leader count gives a sense of the convergence/dominance of the learning as well how cherry-picked the results may be based on number of training epochs. See Fig. 8 for sample training/validation curves which illustrate the advantage of noncompact representations and the competitiveness of our approaches.

### G.2.2 REPRESENTATION TIMINGS

| | Euler | Quat | GS | QCQP | SVD | 2-vec | QMAlg | QMSVD |
|---|---|---|---|---|---|---|---|---|
| Training | 0.2123 | 0.0691 | 0.4903 | 0.5223 | 0.4904 | 0.4447 | 1.2231 | 1.6247 |
| Inference | 0.0401 | 0.0056 | 0.1050 | 0.2435 | 0.2737 | 0.0803 | 0.4298 | 0.6221 |

Table 7: Comparison of timings of different representations run with batch size 128. Measured on Apple M1 Silicon CPU. Values reported are median measurements of 10000 runs in milleseconds. Training includes forward and backward passes (PyTorch train mode), and Inference includes only forward pass (PyTorch eval mode).

Table 7 shows the compute timings of the representations. 2-vec has notably fast inference timings. QuadMobius representations are slower than others as they involve complex arithmetic and more compute steps overall. However, training time differences were observed to be negligible between them and QCQP/SVD as bottlenecks are typically present elsewhere in the pipeline (e.g. data loading, network compute).

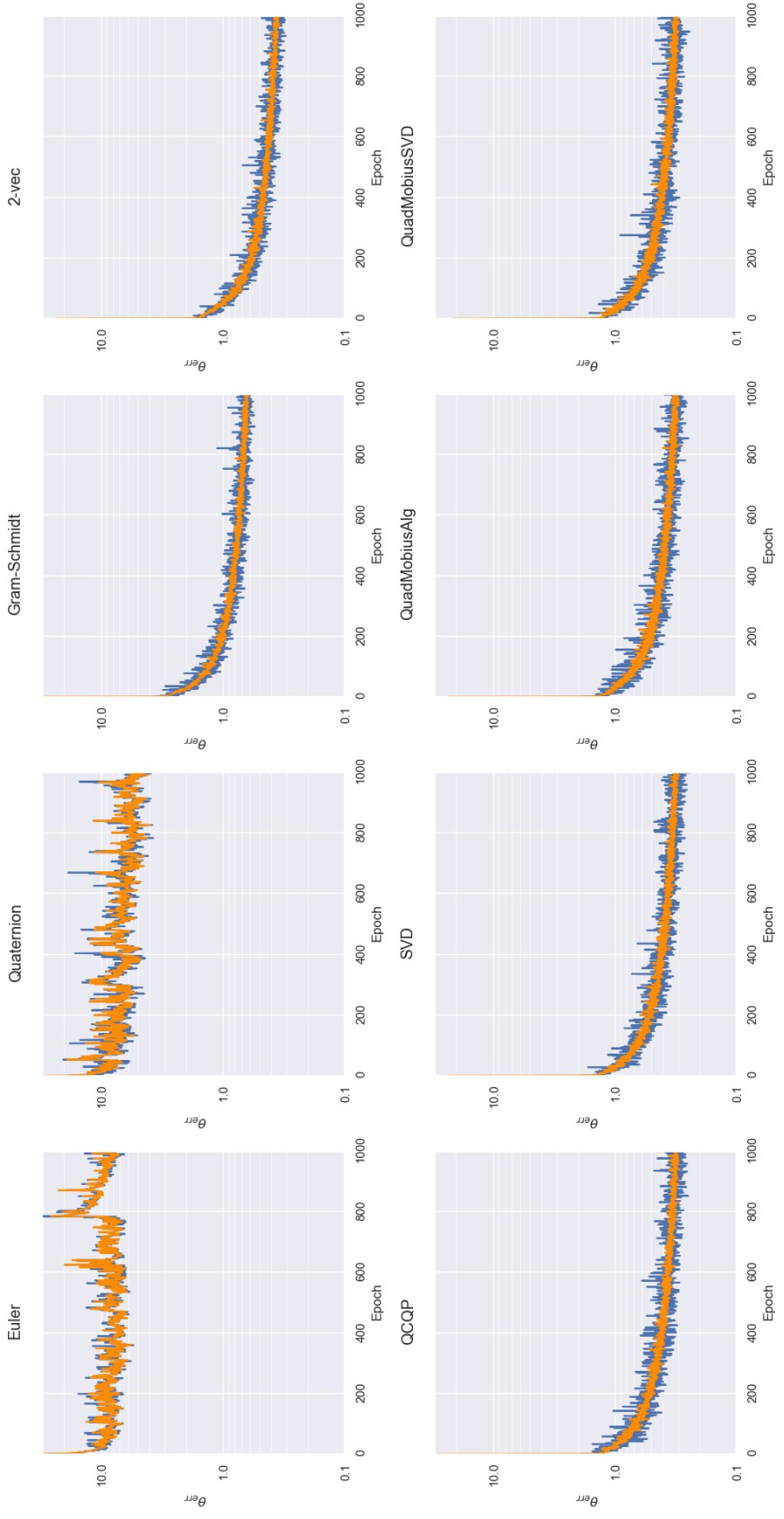

Figure 8: Progression of average $\theta_{err}$ over the training and validation sets for learning Wahba's problem (Appendix G.2.1) for trial #15 in Table 6. Orange is training, blue is validation.

