# OpenReview forum: "Special Unitary Parameterized Estimators of Rotation"
_ICLR.cc/2026/Conference — ICLR 2026 Poster_

### Official Review · Reviewer_gJ8i · 2025-10-31

**Soundness:** 3
**Presentation:** 4
**Contribution:** 3
**Rating:** 8
**Confidence:** 2

**Summary:**

The paper, building on the Wahba problem, proposes two novel representations for learning rotations in neural networks. The first, 2vec, closely resembles the widely used GSO implementation, which has become a standard in neural network training. This representation retains all the desirable properties of the GSO approach, including smoothness and continuity. The second, QuadMobius, introduces a 16-dimensional parameterization for rotations, offering an alternative approach for representing rotational transformations in neural networks.

**Strengths:**

Solid paper - good visualization.
well motivated and good structure to follow along.
coherent notation throughout the paper

**Weaknesses:**

Experimental results show, as expected, no drastic benefit.
Discussion of timing results in the main paper can be emphasized more.

**Questions:**

no questions

---

> ### Author Response · Authors · 2025-11-27
>
> Thank you for your time and positive assessment of our paper! We respond to the comments below.
>
> > Experimental results show, as expected, no drastic benefit. Discussion of timing results in the main paper can be emphasized more.
>
> Regarding performance, we note that our 2-vec representation consistently outperforms the Gram–Schmidt (GS) method (the noted baseline standard) despite sharing the same dimensionality and similar computational cost. As shown in Table 6, 2-vec achieves a lower error in 21 of 24 configurations, at times by a substantial margin (e.g. experiment 15 where GS produces ~80% higher error). Similar trends appear in Fig. 2 and the majority of scenes in Tab. 1 and 2. We attribute this advantage to the more balanced gradient behavior illustrated in Fig. 3 (Appendix F). In addition, QuadMobius achieves the strongest overall results across tasks, including significant gains in several complex-valued experiments in Tab. 6, suggesting a promising advantage in that domain. Beyond these empirical findings, we also contribute new theoretical results (closed-form derivations, proofs, and controlled studies) that underpin the proposed representations.
>
> Regarding the inclusion of timings, we appreciate the feedback. As mentioned in our response to reviewer Xg3U, the QuadMobius timings in Tab. 7 may be further reduced with tailored eigendecomposition routines. At the same time, Tab. 7 also highlights the computational efficiency of 2-vec, which is the fastest smooth representation evaluated. We will strive to bring attention to these observations in the final version.

---

### Official Review · Reviewer_STif · 2025-10-31

**Soundness:** 4
**Presentation:** 2
**Contribution:** 4
**Rating:** 8
**Confidence:** 4

**Summary:**

The paper develops three new representations of rotations based on unitary matrices.

Section 1 introduces Wahba's problem (aka estimate a rotation matrix from vector observations) and provides a brief overview on recent works.

Section 2 first discusses how to measure distances in complex projective space and uses the derived distance metric to formulate Wahba's problem in terms of unitary rotations. This allows the problem to be solved in terms of a set of linear constraints on unitary matrix parameters / quaternions (Section 2.1). Section 2.2 approximates the solution to Wahba's problem by estimating an optimal Möbius transform (a special case of unitary matrix). Section 2.3 derives another solution in terms of linear constraints on quaternions parameters.

Section 3 discusses how the aforementioned solution to Wahba's problem extend to related rotation estimation tasks.

Section 4 proposes to novel formulation of rotations: 2-vec (6D vector that is mapped to an optimal rotation using unitary matrices) and QuadMobius (16D vector is mapped to a Möbius transform and subsequently rotation matrix). QuadMobius is computed either via the SVD or through an algebraic method.

Section 5 compares the proposed rotation representations to other representations (Euler, Quaternions, SVD) on various rotation estimation experiments.

**Strengths:**

Learning with 3D rotations affects numerous applications of machine learning, such that **providing three potentially state-of-the-art rotation representations** is a significant achievement for the ICLR community. The mathematical proofs are elegant and the experiments provide sufficient evidence on the utility of the derived representations. The code appears sound and will be (hopefully) made open-source.

**Weaknesses:**

The writing style of the paper is excellent. However, I personally found the paper's structure to be confusing.

In particular, important prerequisite for understanding the paper are placed in the appendix, while other seemingly unrelated topics are discussed in the main paper:
1) The mathematical tools for this paper are given in Appendix A. Here, I would have liked a 2D illustration of the projective mapping. Also the notation in Equation 30 is a bit confusing. Perhaps, given the projection in (28) and (30) different symbols. Also an overview on the transformations between spaces (SO(3), S^2, CP1, S^3/Quaternion space, Möbius transform space) and the constraints in each space could be insightful.
2) In Section 1, the reader is introduced to Wahba's problem and possible routes to solving it using unitary rotations. Some proofs are given in the section while others are moved into Appendix B.
3) Section 3 talks about solving weighted and unweighted version of Wahba's problem. However, the derived equations are not used in the rest of the paper / experiments, such that I wondered what the purpose of this section was. It seemed this section could have been a paper of its own.
4) Section 4 is somewhat comprehensive, but unfortunately the authors only very briefly mention (around line 350) that the derived representations relate to prior work on learning with rotations. I would have preferred if the authors would have moved derivations to the appendix and shorten/remove Section 3 to instead talk in depth about connections to related work.

**Questions:**

- Line 426 - Is this really an  "Unsupervised learning task"? Seems quite supervised to me as joint locations are given.

---

> ### Author Response · Authors · 2025-11-27
>
> Thank you for your time, thoughtful feedback, and appreciation of our work! We respond to the comments below.
>
> > The mathematical tools for this paper are given in Appendix A. Here, I would have liked a 2D illustration of the projective mapping. Also the notation in Equation 30 is a bit confusing. Perhaps, given the projection in (28) and (30) different symbols. Also an overview on the transformations between spaces (SO(3), S^2, CP1, S^3/Quaternion space, Möbius transform space) and the constraints in each space could be insightful.
>
> Thank you for the pointed feedback. We will strive to improve the clarity of our presentation of formulas, and aim to include such an illustration in our final version.
>
> > In Section 1, the reader is introduced to Wahba's problem and possible routes to solving it using unitary rotations. Some proofs are given in the section while others are moved into Appendix B.
>
> Thank you for the comment. The derivations in Sec. 2 were intended to motivate the connection between Wahba’s problem and special unitary matrices, establishing the unifying framework for the paper. The remaining proofs are more technical and less central to the paper’s narrative, so they were placed in Appendix B.
>
> > Section 3 talks about solving weighted and unweighted version of Wahba's problem. However, the derived equations are not used in the rest of the paper / experiments, such that I wondered what the purpose of this section was. It seemed this section could have been a paper of its own.
>
> We appreciate the comment. The weighted and unweighted two-point solutions were included primarily to provide the theoretical motivation and derivation for the 2-vec representation, connecting it to the broader conceptual framework established earlier. For instance, the proof for the unweighted case (which underlies 2-vec) borrows from the proof for the weighted case. To connect the theory with practice, the solutions are also benchmarked and compared in Tab. 5 (Appendix G) in the same problem context as our other methods (Tab. 4, Tab. 6), with code implementations provided in the Suppl. Mat. While we hope this section serves as a theoretical foundation for future applications, we will strive for an improved content structure to better reinforce the paper’s narrative.
>
> > Section 4 is somewhat comprehensive, but unfortunately the authors only very briefly mention (around line 350) that the derived representations relate to prior work on learning with rotations. I would have preferred if the authors would have moved derivations to the appendix and shorten/remove Section 3 to instead talk in depth about connections to related work.
>
> Thank you for the helpful feedback. We agree that strengthening the discussion of connections to prior work would improve context for readers, and we will look for opportunities to expand this section and add further intuition behind our proposed learning methods in the final version.
>
> > Line 426 - Is this really an "Unsupervised learning task"? Seems quite supervised to me as joint locations are given.
>
> Thank you for pointing this out. The task is indeed supervised with respect to joint locations. Our intention was to convey the fact that the rotations themselves are not explicitly supervised (as is the case in every other experiment). Perhaps “implicit learning” may be a better term.

---

### Official Review · Reviewer_JA2J · 2025-11-01

**Soundness:** 3
**Presentation:** 3
**Contribution:** 3
**Rating:** 8
**Confidence:** 2

**Summary:**

This paper proposes new parameterization methods for 3D rotation estimators. The method is based on operation in the complex number domain, and 3 methods are proposed including 2-vector, QMAlg and QMSVD. The proposed method is benchmarked on Wahba's problem, point cloud pose estimation, inverse Kinematics and camera pose estimation.

**Strengths:**

- The parameterization of rotation estimator is a challenging and practically important topic in machine learning.
- The paper introduce new theoretical results and their proofs.
- The experimental results are extensive and the results are mostly better than the baselines.

**Weaknesses:**

- The paper is math heavy and may not be easy for general reader to get a gist of the approach.

**Questions:**

- Do the proposed approaches solve the discontinuity and double cover problems of 3D rotation parameterization?
- Do the baseline methods (e.g., GS) also solve these two problems? The GS baseline seems to work well for point cloud data and camera pose data, I am curious whether the importance of solving discontinuity or/and double cover issues are shown in the results.
- What are the numbers of output elements need for the proposed methods to parametrize a rotation?
- Figure 1 provides some intuitive illustration of the method. For QuadMobius, is there an interpretation for the fact that projecting to intermediate points can improve the results?

---

> ### Author Response · Authors · 2025-11-27
>
> Thank you for your time, accurate summary of our contributions, and thoughtful questions! We answer your questions below.
> > The paper is math heavy and may not be easy for general reader to get a gist of the approach.
>
> Thank you for the feedback. We will strive for improved clarity and intuition behind our equations and proofs in a final version.
>
> > Do the proposed approaches solve the discontinuity and double cover problems of 3D rotation parameterization?
>
> Yes, both 2-vec and QuadMobius avoid those issues, allowing for smooth learning in the context of `Zhou et al (2019)`, while quaternion and Euler angle representations do not. This difference is reflected in the large performance gaps in nearly every benchmark (Tab. 1, table in Fig. 2, Tab. 6, Fig. 7), highlighting the importance of continuity in learning representations.
>
> > Do the baseline methods (e.g., GS) also solve these two problems? The GS baseline seems to work well for point cloud data and camera pose data, I am curious whether the importance of solving discontinuity or/and double cover issues are shown in the results.
>
> Gram-Schmidt also provides a smooth representation, thereby avoiding the aforementioned topological issues. However, our experiments show that it consistently performs weaker than the proposed methods. This difference is most evident in Tab. 6 and Fig. 7 where GS often converges to a substantially higher error than 2-vec and QuadMobius (in some cases, like experiment 15, more than twice as large). The same behavior appears in Tab. 1 (Chair/Toilet) and in the percentile comparison in Fig. 2. Even in settings where GS is relatively competitive (Sofa in Tab. 1 or all scenes in Tab. 3), the proposed methods remain the dominant performers. These experimental results are supported by Fig. 3 (Appendix F) which shows that GS produces a more imbalanced gradient flow relative to 2-vec, suggesting less stability in optimization.
>
> > What are the numbers of output elements need for the proposed methods to parametrize a rotation?
>
> 2-vec parameterizes two 3D axes, requiring 6 parameters (the same dimensionality as GS). QuadMobius parameterizes a $4\times 4$ Hermitian matrix, requiring 16 unique real parameters.
>
> > Figure 1 provides some intuitive illustration of the method. For QuadMobius, is there an interpretation for the fact that projecting to intermediate points can improve the results?
>
> In the context of learning rotations, an effective representation should both provide a strong learning signal and remain stable under noisy inputs. These goals often oppose one another, and existing methods attempt to satisfy both through a single projection step. QuadMobius instead proposes a two-stage process that separates the roles. The eigendecomposition step helps buffer against noise (analogous to a least-squares estimate), creating a stable latent representation that we identify with a Möbius transformation in the context of Sec. 2.2. The subsequent projection to $SU(2)$ serves as a structured correction step, establishing a strong connection between the latent space and rotational manifold. In this way, the two parts serve distinct purposes while working in tandem. This interpretation is supported by the theoretical investigations in Tab. 3, Fig. 4, and Fig. 5 in Appendix F and motivated by the context of Wahba’s problem in Sec. 2.2.

---

### Official Review · Reviewer_Xg3U · 2025-11-04

**Soundness:** 3
**Presentation:** 4
**Contribution:** 4
**Rating:** 10
**Confidence:** 4

**Summary:**

This work presents a comprehensive and theoretically grounded exploration of 3D rotation estimation, introducing Wahba's problem's solution via the special unitary group SU(2). This work proposes two continuous representations for learning rotations in deep neural networks:
1. 2-vec: different from Gram-Schmidt's greedy approach, 2-vec maps these vectors to an optimal rotation in the sense of Wahba's problem.
2. QuadMobius:  A 16D representation based on the paper's Möbius transformation approximation. It maps the network output to a Hermitian matrix, solves for its smallest eigenvector, and then projects onto SU(2) to get the final rotation.
The authors demonstrate these contributions through extensive experiments, including 3D shape alignment, unsupervised inverse kinematics, and real-world camera pose estimation.

**Strengths:**

1. The reformulation of Wahba’s problem via SU(2) presents a rigorous mathematical derivation. Translating theoretical derivation to practical representation (2-vec and QuadMobius) of deep rotatiton estimation is novel and intriguing.
2. The paper tests its representations on a diverse set of tasks on ModelNet10-SO3, Inverse Kinematics,  and Cambridge Landmarks, providing strong evidence of their robustness and general effectiveness of the proposed representation.

**Weaknesses:**

1. In Appendix F, the author tried to directly predit an SU(2) form but it works poorly due to the double cover issue like quaternions. It raises the question that the true improvement stems from the intermediate Möbius transformation rather than the property from SU(2). The paper might want to clarify its narrative to highlight the key contributing factor.
2. The QuadMobius has noticeably slower inference speed, shown in Tab. 7, but I believe this is a minor issue as running the neural network inference would dominate the slight increase of arithmetic computation from QuadMobius.

**Questions:**

I suggest the authors highlight the intuition why SU(2) derived rotation form performs better early on in the paper. Though the theoretical in-depth derivation is presented beautifully, it's unclear whether the true source of improvement of QuadMobius is from the properties of SU(2) or Mobius transformation. I hope the authors could clarify this.

---

> ### Author Response · Authors · 2025-11-27
>
> Thank you for your time, pointed feedback, and appreciation of our work! We address the comments below.
>
> >In Appendix F, the author tried to directly predit an SU(2) form but it works poorly due to the double cover issue like quaternions. It raises the question that the true improvement stems from the intermediate Möbius transformation rather than the property from SU(2). The paper might want to clarify its narrative to highlight the key contributing factor.
>
> Thank you for the insightful comment. We can clarify the intuition behind the roles of $SU(2)$ projection and eigendecomposition for QuadMobius. Since $SU(2)$ is isomorphic to quaternions, its form can be considered redundant in a classical setting (the bottom row repeats the information in the top row). However, this redundancy becomes beneficial in a learning context. The network can implicitly form two quaternion-like hypotheses before projection, and the projection step then provides an elegant, structured method to aggregate the information into a single rotation. In practice, this tends to make $SU(2)$ projection more performant than predicting quaternions directly. However, $SU(2)$ alone fails to be a top performer since it also suffers from the double-cover ambiguity and can be sensitive to noise during training. The eigendecomposition step naturally resolves both issues by empirically creating a buffered space for projection to learn from and guarding against arbitrary scaling during optimization. Yet, without the structured guidance provided by projecting to $SU(2)$, eigendecomposition by itself does not consistently produce a strong learning signal. These interpretations are supported by our theoretical experiments in Tab. 3, Fig. 4, and Fig. 5 in Appendix F.
>
> >The QuadMobius has noticeably slower inference speed, shown in Tab. 7, but I believe this is a minor issue as running the neural network inference would dominate the slight increase of arithmetic computation from QuadMobius.
>
> Thank you for the observation. We agree that the inference overhead for QuadMobius is typically minor relative to the overall runtime. We note that much of its computational cost comes from the eigendecomposition step (using `torch.linalg.eigh` in PyTorch). This operation is slower than necessary for our setting because it computes a full eigendecomposition, synchronizes with the CPU on CUDA backends, and relies on complex-valued routines that are generally less optimized than their real-valued counterparts. Since our method only requires the eigenvector associated with the smallest eigenvalue, a tailored routine (e.g. shifted inverse iteration) could likely reduce runtime in practice. On the other hand, our 2-vec representation already offers a computationally lightweight alternative as evidenced by Tab. 7.

---

### Author Response · Authors · 2025-11-23
**Thank You**

We sincerely thank all the reviewers for the time and effort spent evaluating our work and for the thoughtful responses provided. We are humbled by your recognition of our work, and we look forward to responding to your comments to discuss it further!

Also, a big thank you to all the chairs involved for their time in ensuring the review process goes smoothly, especially given the record number of submissions this year. Your work is quite valuable to the ICLR community!

---

### Author Response · Authors · 2025-12-03
**First Revision**

Thank you once again to all the reviewers for their valuable feedback in helping us to strengthen our paper! We have implemented the following changes already:

- added illustration of stereographic projection and clarified projection notation as per reviewer STif
- added figure visualizing the dominant performance of 2-vec over Gram-Schmidt as per reviewers JA2J, gJ8i
- minor typo fixes and cleaning up of sections

and we aim to revise our paper in a final version in the following ways based on the feedback:
- discuss in more depth the connection of our learning representations to existing representations as per reviewer STif
- further explain the intuition of our representations as well as interpret the role each step of our projections plays as per reviewers Xg3U, JA2J
- discuss the timing results of our learning representations in the main paper body as per reviewers Xg3U, gJ8i

---

### Meta-Review · Area_Chair_xJdk · 2026-01-05

**Summary:**

In this study, the authors revisit the rotation estimation problem, reformulating Wahba's problem as a quaternion-based constrained optimization problem. This problem can be solved efficiently through eigen-decomposition. Moreover, it provides a new way to represent and learn rotations based on neural networks. Experiments on solving Wahba's problem and camera pose estimation demonstrate the feasibility of the proposed method. All the reviewers admit the solidity of the proposed method.

**Reviewer Concerns:**

The main concerns are about the readability of the submission. In the rebuttal phase, the authors made some efforts to address the readability issues, but, in my opinion, the paper's writing can be further improved. For example, each equation in the submission should be a part of corresponding sentences, but none of the equations have punctuation marks. In addition, some equations shown in Appendix are cited in the main paper, which increases the reading difficulty and breaks the self-containness of the main paper. I strongly suggest the authors further polish and revise the paper.

**Reviewer Scores:**

I believe the reviewers would have maintained their scores if they had been able to discuss with the authors.

---

### Decision · Program_Chairs · 2026-01-26

Accept (Poster)